# TrajP-L: A Trajectory-Based Plugin with LoRA for Sampling Direction Correction in Distilled Diffusion Models

## Abstract

Diffusion models (DMs) have shown remarkable capability in image synthesis, yet they typically require hundreds of sampling steps to produce high-quality outputs. To alleviate this inefficiency, prior work has explored distilling the sampling trajectories of pre-trained models. However, these approaches often disrupt the original parameter space and incur substantial distillation training costs. Recent findings suggest that the sampling space of DMs can be effectively captured by as few as three basis vectors, with the resulting low-dimensional trajectories exhibiting strong structural similarity. Based on this insight, we propose TrajP-L (Trajectory-Based Plugin with LoRA), a trajectory similarity-based learnable plugin. It achieves efficient DMs distillation via the synergy of LoRA and a trajectory correction module (TrajP). Specifically, we construct a student model by combining LoRA with the weights of a pre-trained model to initialize the distillation process. We then extract the coordinate information of the current and next sampling timesteps from a fitted 3D trajectory representation, and employ TrajP to refine the student's sampling direction. Extensive experiments show that TrajP-L requires only a small number of sampling trajectories for fine-tuning, while substantially mitigating discretization errors. For example, on CIFAR-10, TrajP-L trains on merely 5k trajectories for 10 minutes on a single NVIDIA RTX 3090 GPU, improving DDIM performance from 169.50 FID (NFE=2) to 5.02.

## 1 Introduction

Generative models are a class of deep learning models that learn the underlying probability distribution of data and generate samples resembling the training set. Representative approaches include Generative Adversarial Networks (GANs) (Goodfellow et al., 2020), Variational Autoencoders (VAEs) (Kingma & Welling, 2013), and Diffusion Models (DMs) (Sohl-Dickstein et al., 2015; Song et al., 2020a; Ho et al., 2020). Among them, DMs stand out for their forward noising and reverse denoising diffusion structure and have achieved substantial progress in a wide range of tasks, such as image generation (Sohl-Dickstein et al., 2015; Song & Ermon, 2019; Song et al., 2020b), text-to-image generation (Rombach et al., 2022), video generation (Zheng et al., 2024), text-to-3D generation (Poole et al., 2022), and audio generation (Evans et al., 2024). However, due to the inherent limitations of the diffusion process, DMs typically require dozens to hundreds of sampling steps to produce high-quality results. The slow sampling speed and large computational overhead significantly hinder their practical deployment.

The acceleration of diffusion model sampling can be broadly categorized into two families: solver-based methods (Karras et al., 2022; Liu et al., 2022; Lu et al., 2022; 2025; Song et al., 2020a; Zhang & Chen, 2022; Zhao et al., 2023; Zheng et al., 2023; Wang et al., 2024a; Zhou et al., 2024a; Zhu et al., 2025) and distillation-based methods (Salimans & Ho, 2022; Yin et al., 2024b; Zhou et al., 2024b; Wang et al., 2024b). Solver-based methods focus on designing fast solvers to reduce the discretization error of individual steps, and can generate images of comparable quality to the original 1000-step process with only a dozen denoising iterations. However, due to the inherent accumulation of discretization errors, their performance deteriorates sharply when the number of function evaluation (NFE) drops below 10, leading to a significant loss in sample quality. To overcome this limitation, a substantial body of research has focused on distillation strategies for pre-trained

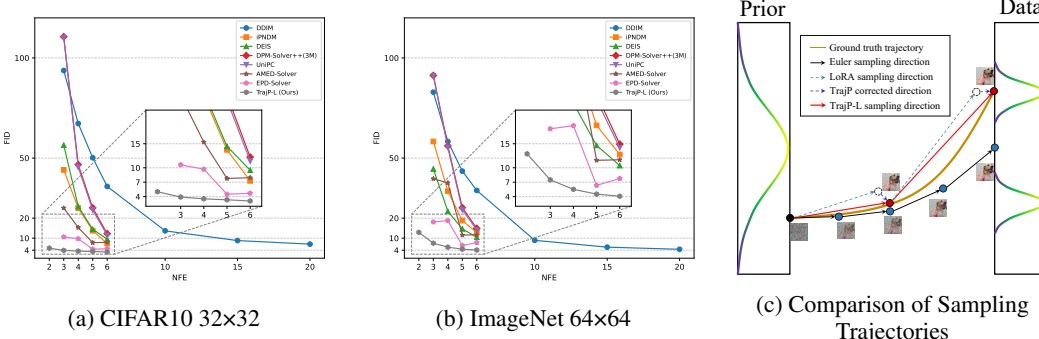

(a) CIFAR10 32×32      (b) ImageNet 64×64      (c) Comparison of Sampling Trajectories

Figure 1: (a) Visual Comparison on CIFAR10 32×32 (Krizhevsky & Hinton, 2009) based on FID↓. (b) Visual Comparison on FFHQ 64×64 (Karras et al., 2019) dataset based on FID↓. (c) Comparison of partial sampling trajectories between TrajP-L and a first-order Euler solver (Song et al., 2020a), where the update directions are guided by the tangent directions of the sampling trajectories.

DMs. (Luhman & Luhman, 2021; Luo et al., 2023a; Berthelot et al., 2023). Distillation-based methods are generally more effective than solvers in low-NFE regimes, and some even establish a direct mapping between the noise and data distributions, enabling one-step generation of images with quality comparable to the full 1000-step process (Song et al., 2023; Kim et al., 2023; Yin et al., 2024b). Nevertheless, these approaches suffer from a critical drawback: as the number of sampling steps decreases, the training cost of the student model grows substantially—sometimes even exceeding that of the teacher model—severely limiting practical deployment. To mitigate this issue, recent studies have attempted to distill entire sampling trajectories and have achieved some progress (Zhou et al., 2024b; Wang et al., 2024b). Yet, these methods either require an extremely large number of trajectories ($\geq$ 200K) or perform poorly under very low NFE ($\leq$ 5) settings (Wang et al., 2024b).

To further reduce the distillation cost of DMs, we revisit the role of sampling trajectories in the distillation process. Existing approaches often rely on constructing complete teacher trajectories, which makes the process computationally expensive. However, our study reveals that DMs trajectories exhibit strong structural similarity not only in low-dimensional spaces (Chen et al., 2024), but also in the original sampling space (see Appendix C.3). This key observation motivates us to leverage trajectory-shape information together with only a small number of teacher trajectories, to guide the student model. By correcting the student's sampling direction with this trajectory-aware prior, discretization errors during few-step sampling with large intervals can be significantly mitigated. Based on this idea, we propose TrajP-L, a trajectory-based plugin with LoRA, to construct the student model and further refine its denoising direction using the three-dimensional geometry of teacher trajectories. Experiments demonstrate that TrajP-L not only accelerates the distillation process but also substantially reduces discretization errors, enabling a practical and efficient paradigm of "trajectory-aware distillation" for DMs.

The key contributions of this paper are summarized as follows:

- We verified the strong similarity of diffusion model sampling trajectories in the original sampling space and fitted a mathematical expression of the trajectories in three-dimensional space, enabling precise acquisition of trajectory information at any timestep;

- We proposed the trajectory correction plugin TrajP-L, which leverages the fitted trajectory expression to dynamically correct the denoising direction at each sampling step, effectively reducing the cumulative error of DMs under few-step sampling with large intervals;

- We conducted extensive experiments on multiple datasets, and the results show that the proposed TrajP-L plugin can effectively accelerate the sampling speed of DMs while introducing virtually no additional computational overhead.

## 2 PRELIMINARIES

### 2.1 DIFFUSION AND DENOISING PROCESSES

DMs (Sohl-Dickstein et al., 2015; Ho et al., 2020; Song et al., 2020b) learn a mapping between data and noise distributions through a forward diffusion process and a reverse denoising process.

#### 2.1.1 FORWARD DIFFUSION

Given $x_0$ and the current time step $t$, where $x_0 \sim P_{\text{data}}(x)$ and $t \in [0, T]$, the probability distribution $q(x_t|x_0)$ of $x_t$ can be obtained by adding standard Gaussian noise $\epsilon$ of varying intensities to $x_0$ at different time steps $t$:

$$x_t = \alpha_t x_0 + \sigma_t \epsilon,$$
$$q(x_t|x_0) = \mathcal{N}(x_t; \alpha_t x_0, \sigma_t^2 I). \tag{1}$$

Here, both $\alpha_t$ and $\sigma_t$ are functions of $t$. When representing $x_t$ using a forward continuous SDE (Song et al., 2020b), we have:

$$dx_t = f(t)x_t dt + g(t)dw_t. \tag{2}$$

Combining with Eq. (1), we get $f(t) = \frac{d}{dt}(\ln \alpha_t) = \frac{1}{\alpha_t}\frac{d\alpha_t}{dt}$, $g(t)^2 = \alpha_t^2 \frac{d}{dt}\left(\frac{\sigma_t^2}{\alpha_t^2}\right) = 2\alpha_t \sigma_t \frac{d}{dt}\left(\frac{\sigma_t}{\alpha_t}\right)$, and $w_t \in \mathbb{R}^D$ is a standard Wiener process.

#### 2.1.2 REVERSE DENOISING PROCESS

DMs map the prior distribution back to the target distribution through reverse denoising process. The reverse denoising process can be expressed as the inverse of Eq. (1) and Eq. (2). Since different values of $\alpha_t$ and $\sigma_t$ in Eq. (1) can be represented by Eq. (2), this paper mainly discusses the reverse denoising process in the form of SDE. According to research on Song et al. (2020b), there exists an equivalent reverse SDE diffusion process for Eq. (2):

$$dx_t = [f(t)x_t - g^2(t)\nabla_x \log q_t(x_t)]dt + g(t)d\overline{w}_t. \tag{3}$$

Here, $\overline{w}_t$ denotes a standard Wiener process with time flowing backward from $T$ to $0$. The term $\nabla_x \log q_t(x_t)$, referred to as the score function, represents the gradient of the log-density of the data distribution $q_t(x_t)$ at $x_t$, and characterizes both the direction and the strength of denoising from the noisy sample $x_t$ toward the clean sample $x_0$. Since $\nabla_x \log q_t(x_t)$ is generally intractable, it is approximated using neural networks. To remove the randomness in Eq. (3), Song et al. (2020b) derived the probability flow ordinary differential equation (PF-ODE), which shares the same marginal distribution as Eq. (3) at any time $t$, based on the Fokker–Planck equation, as follows:

$$dx_t = [f(t)x_t - \frac{1}{2}g^2(t)\nabla_x \log q_t(x_t)]dt. \tag{4}$$

The reverse denoising process represented by Eq. (4) eliminates the influence of random noise, and its solution process is more stable and computationally efficient. Samples can be generated step by step through deterministic ODE solvers (such as Euler's method (Song et al., 2020a), Runge-Kutta method (Liu et al., 2022)), thus having higher practical value.

### 2.2 SCORE MATCHING

To solve the reverse denoising process of Eq. (4), Song & Ermon (2019) introduced the score matching method, using a neural network $s_\theta(x_t, t)$ to estimate the logarithmic density gradient $\nabla_x \log q_t(x_t)$ at $x_t$ at any time step $t$. Song & Ermon (2019) pointed out that $\nabla_{x_t} \log p(x_t|x_0)$ can perfectly replace $\nabla_x \log p_t(x)$, so the loss function of the neural network can be expressed as:

$$\mathbb{E}_{x_0 \sim q_{\text{data}}} \mathbb{E}_{x_t \sim q(x_t|x_0)} \|s_\theta(x_t, t) - \nabla_{\boldsymbol{x}_t} \log p(\boldsymbol{x}_t|\boldsymbol{x}_0)\|_2^2. \tag{5}$$

Some studies use a noise prediction model $\epsilon_\theta(x_t, t)$ to estimate the Gaussian noise added to $x_0$ (Ho et al., 2020; Nichol & Dhariwal, 2021), while others (Salimans & Ho, 2022) use a data prediction model $D_\theta(x_t, t)$ to reconstruct $x_0$ from $x_t$. Both are closely related to the score-based model $s_\theta(x_t, t)$, with relationships expressed as follows:

$$s_\theta(x_t, t) = -\frac{\epsilon_\theta(x_t, t)}{\sigma_t} = \frac{D_\theta(x_t, t) - x_t}{\sigma_t^2}. \tag{6}$$

We follow the setting of EDM (Karras et al., 2022), letting $f(t) = 0$, $g(t) = \sqrt{2t}$, $\sigma(t) = t$, and substitute the learned score function Eq. (6) into Eq. (4) to obtain a simplified form of PF-ODE:

$$dx = -s_\theta(x_t, t)tdt = \epsilon_\theta(x_t, t)dt. \tag{7}$$

The exact solution of Eq. (7) is:

$$x_{t_{i-1}} = x_{t_i} - \int_{t_i}^{t_{i-1}} s_\theta(x_t, t)tdt = x_{t_i} + \int_{t_i}^{t_{i-1}} \epsilon_\theta(x_t, t)\, dt. \tag{8}$$

## 3 METHOD

### 3.1 FITTING OF SAMPLING TRAJECTORIES IN THREE-DIMENSIONAL SPACE

To fully leverage the similarity of diffusion model sampling trajectories for accelerating the distillation process, in this section we attempt to fit a mathematical representation of the sampling trajectories in three-dimensional space, enabling the extraction of trajectory coordinates at arbitrary timesteps. The extracted trajectory coordinates can then be incorporated as prior information into TrajP-L to correct the sampling direction, ensuring that it remains closely aligned with the teacher's sampling trajectories. We first visualized the shapes of the first three principal components of the sampling trajectories in the CIFAR10 32×32 (Krizhevsky & Hinton, 2009) respectively, as shown in Figure 2. We found that the shapes of these principal components basically conform to the properties of polynomial, exponential, or Gamma functions. Additionally, we attempted to use Fourier functions as the basis functions, but the results were not satisfactory, and the relevant visualization results are presented in Figure 3.

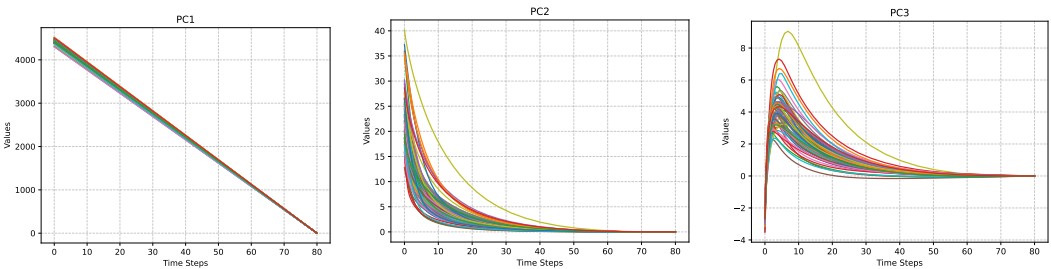

Figure 2: Visualization results of the shape of the first three principal components of sampling trajectories for the CIFAR10 32×32 (Krizhevsky & Hinton, 2009).

### 3.1.1 IMPLEMENTATION OF TRAJECTORY FITTING

We begin by sampling 64 trajectories from the pre-trained model and fitting their PCA-transformed representations. Unlike (Chen et al., 2024), we designate the direction of $x_{t_{N-1}} - x_{t_N}$ as the first principal component, since $x_{t_{N-1}}$ can be readily obtained during sampling. The output of each dimension (corresponding to $PC_1, PC_2, PC_3$) is modeled as the combination of three components: a quadratic polynomial term, an exponential term, and a Gamma term. At time step $t$, the value of the $k$-th dimension ($k = 1, 2, 3$) is given by:

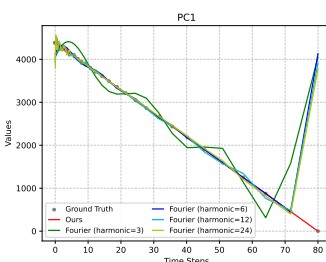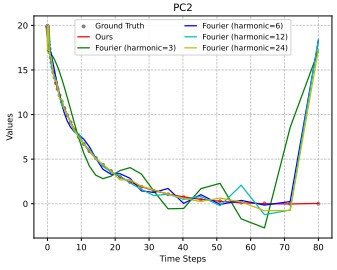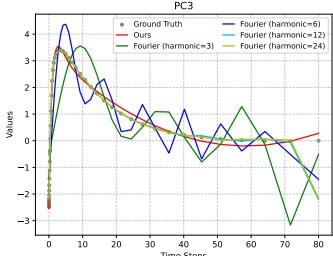

Figure 3: Visualization results of polynomial, exponential and gamma basis functions versus fourier basis functions on the first three principal components of sampling trajectories in the CIFAR10 32×32 (Krizhevsky & Hinton, 2009).

$$\text{output}_k(t) = \text{quadratic}_k(t) + \text{exponential}_k(t) + \text{gamma}_k(t). \tag{9}$$

The specific expressions of the three components are as follows:

**Quadratic Polynomial Term**:

$$\text{quadratic}_k(t) = P_k \cdot t^2 + Q_k \cdot t + R_k. \tag{10}$$

**Exponential Term**:

$$\text{exponential}_k(t) = A_k \cdot \exp\left(\lambda_k \cdot t\right). \tag{11}$$

**Gamma Term**:

$$\text{gamma}_k(t) = G_{f_k} \cdot t^{G_{g_k}} \cdot \exp\left(-\frac{t}{G_{h_k}}\right). \tag{12}$$

Where $P_k, Q_k, R_k$ denote the coefficients of the quadratic, linear, and constant terms of the polynomial, respectively. Similarly, $A_k$ and $\lambda_k$ represent the coefficient and the exponential rate of the exponential term, while $G_{f_k}, G_{g_k}, G_{h_k}$ correspond to the coefficient, the power exponent, and the exponential decay parameter of the Gamma term, respectively. In vectorized form, the expression can be written uniformly as:

$$\begin{aligned}
\text{output}(t) &= \left(P\,t^2 + Q\,t + R\right) + \left(A \odot \exp\left(\lambda\,t\right)\right) + \left(G_f \odot t^{G_g} \odot \exp\left(-\frac{t}{G_h}\right)\right) \\
&= [\text{output}_1(t), \text{output}_2(t), \text{output}_3(t)].
\end{aligned} \tag{13}$$

Here, the three dimensions of $\text{output}(t)$ correspond to the coefficients of the three principal components, $PC_1$, $PC_2$, and $PC_3$, at time $t$. The shape of the fitted trajectories is illustrated in Figures 8–9, and the model parameters for different datasets are listed in Tables 10–11. The similarity of diffusion model sampling trajectories is further validated in Appendix C.

### 3.2 LoRA-Based diffusion model distillation mechanism

Parameter-Efficient Fine-Tuning (PEFT) (Han et al., 2024) offers a lightweight way to adapt pre-trained models. Among PEFT methods, LoRA (Hu et al., 2022) reduces computational cost by constraining updates to a low-rank space while preserving performance. Since diffusion model distillation via sampling trajectories is effectively task-specific fine-tuning, LoRA can be directly applied to efficiently update the student model. For any weight matrix $W_0 \in \mathbb{R}^{d \times k}$, LoRA's update via low-rank decomposition is $\Delta W = BA$, and the forward pass for input $x$ is modified accordingly:

$$h = W_0 x + \frac{\alpha}{r} \cdot BAx, \tag{14}$$

where $B \in \mathbb{R}^{d \times r}$ and $A \in \mathbb{R}^{r \times k}$ are low-rank matrices (rank $r \ll \min(d, k)$), $\alpha$ is a scaling factor. The pre-trained weight $W_0$ remains fixed during training, and only $A$ and $B$ need to be optimized to achieve parameter-efficient updates.

In trajectory distillation, the student model aims to match the teacher's multi-step sampling trajectory using a single-step Euler method (see Appendix A). With LoRA, the student parameters $\theta$ comprise the pre-trained weights $W_0$ and low-rank matrices $A_i^\theta, B_i^\theta$ for layers where LoRA is applied. The loss function can be expressed as:

$$\mathcal{L}_\theta(\{A_i^\theta, B_i^\theta\}) = \mathbb{E} \left\| x_{t_{n-1}} - \hat{x}_{t_{n-1}} \right\|. \tag{15}$$

Here, $\hat{x}_{t_{n-1}}$ denotes the denoised output of the student model, while $x_{t_{n-1}}$ denotes that of the teacher model. To reduce the number of trainable parameters and mitigate the risk of overfitting, we introduce trainable LoRA layers into half of the linear and convolutional layers (see Figure 4). The values of $r$ and $\alpha$ are listed in Tables 3, 7, and 15. By leveraging LoRA, the number of trainable parameters in the distillation process is reduced from $\mathcal{O}(dk)$ to $\mathcal{O}(r(d + k))$, achieving over 95% parameter compression (see Table 8).

### 3.3 THE PROPOSED TRAJP-L

---

**Algorithm 1** Correction

---

**Require:** trajectory buffer $Q$, trajectory information $u, v, w$, denoising direction $\hat{d}_t$
1: Obtain $x_{buff} = \{x_N, \tilde{d}_{t_N}, ..., \tilde{d}_{t_{n-1}}\}$ from $Q$      ▷ Obtain the denoising starting point and historical denoising directions.
2: $x_{buff}.\text{append}(\hat{d})$
3: $x_{buff} = concat(x_{buff})$
4: $[PC1, PC2, PC3] = Conv(x_{buff})$      ▷ Implicit PCA via convolution
5: $d_c = u \cdot PC1 + v \cdot PC2 + w \cdot PC3$
6: $\tilde{d}_n = \alpha \cdot \hat{d}_t + (1 - \alpha) \cdot d_c$      ▷ Correct the sampling direction.
7: **return** $\tilde{d}_{t_n}$

---

To further reduce discretization errors under few-step sampling with large intervals and fully exploit the shape similarity of diffusion model trajectories, we propose TrajP-L, a trajectory-based plugin with LoRA. This plugin leverages the trajectory prior fitted in Section 3.1 to dynamically adjust the sampling direction of the student model, significantly improving sampling accuracy without increasing model parameter complexity. The workflow is as follows: based on the fitted trajectory expression output($t$), we compute the coordinate difference between the current time $t_n$ and the target time $t_{n-1}$ in the low-dimensional principal component space:

$$[u, v, w] = \text{output}(t_{n-1}) - \text{output}(t_n). \tag{16}$$

The denoising direction of TrajP-L can be expressed as:

$$\tilde{d}_{t_n} = T_\theta(\hat{d}_{t_n}, Q, u, v, w). \tag{17}$$

Here, $\hat{d}_{t_n}$ is the denoising direction output by pre-trained model with LoRA, and $Q$ is the cached denoising starting point and denoising direction. The denoising output of the next step can be expressed as:

$$\tilde{x}_{t_{n-1}} = x_{t_n} + (t_n - t_{n-1})\tilde{d}_{t_n}. \tag{18}$$

During the training process, TrajP and LoRA are updated through joint optimization, and the loss function is defined as:

$$\mathcal{L}_\theta = \mathbb{E} \left\| x_{t_{n-1}} - \tilde{x}_{t_{n-1}} \right\|. \tag{19}$$

Here, $\tilde{x}_{t_{n-1}}$ denotes the denoised output produced by TrajP-L. By leveraging LoRA for parameter-efficient updates and exploiting the trajectory similarity verified in Section 3.1, TrajP-L requires only 3K–7K trajectories to cover the full sampling space (see Table 16), substantially reducing both data collection and training costs. The detailed correction process is shown in Algorithm 1, and the training and sampling procedures of TrajP-L are detailed in the Algorithms 3–4.

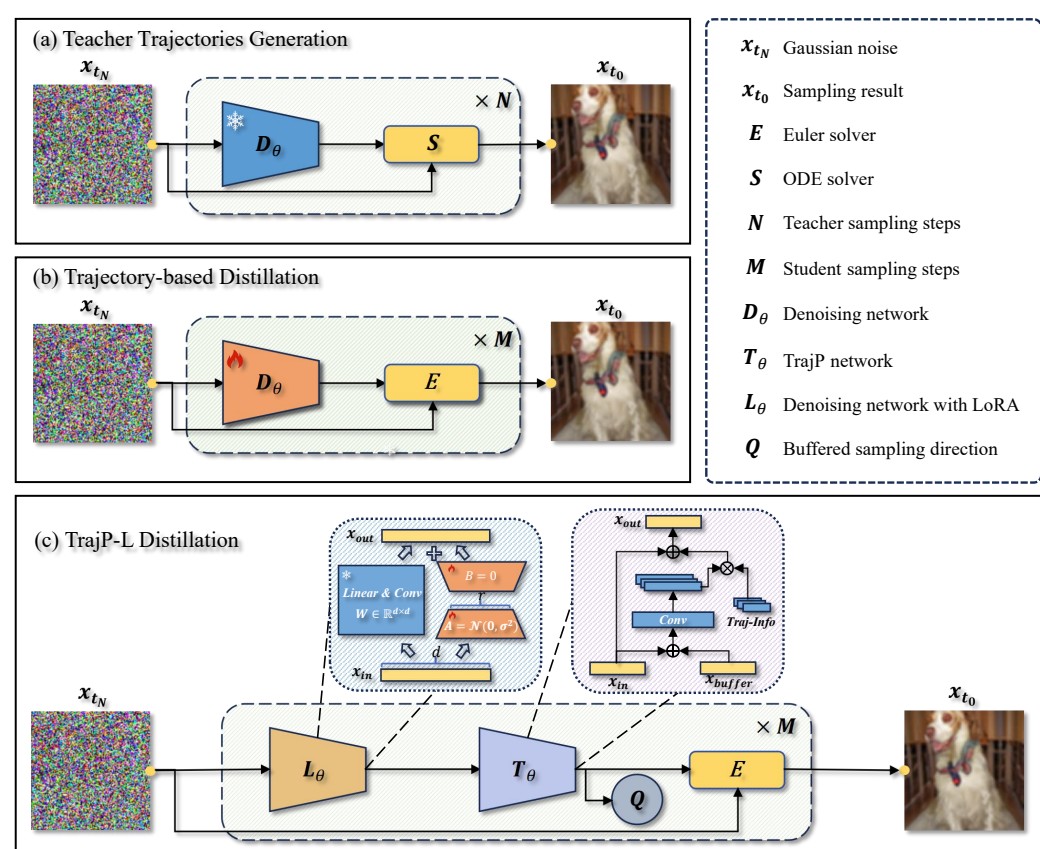

Figure 4: Schematic Diagram of Trajectory Generation and Distillation. (a) We use a pre-trained teacher model $D_\theta$ to generate and save sampling trajectories. For Stable Diffusion v1.5 (Rombach et al., 2022), the solver $S$ adopts DPM-Solver++(2M) (Lu et al., 2025); for other models, the solver $S$ uses DPM-Solver++(3M) (Lu et al., 2025). (b) Other distillation-based methods construct student models by retraining or fine-tuning the entire teacher model to mimic complete sampling trajectories (Song et al., 2023; Zhou et al., 2024b), and the solver $E$ generally adopts DDIM (Song et al., 2020a). (c) TrajP-L introduces LoRA (Hu et al., 2022), which effectively reduces the number of trainable parameters by over 95%. It performs implicit PCA by caching denoising directions and corrects the sampling direction using information from the fitted trajectory function.

## 4 EXPERIMENTS

### 4.1 EXPERIMENT SETTING

**Datasets and pre-trained models.** We employ TrajP-L for model distillation across a wide range of image resolutions (32–512) and spaces (pixel and latent spaces). In the pixel space, we utilize CIFAR10 32×32 (Krizhevsky & Hinton, 2009), FFHQ 64×64 (Karras et al., 2019), and ImageNet 64×64 (Deng et al., 2009). In the latent space, we adopt LSUN-Bedroom 256×256 (Yu et al., 2015) and 512×512 resolution images generated using the Stable Diffusion v1.5 checkpoint (Rombach et al., 2022). Both student and teacher network parameters are initialized from pre-trained diffusion models provided by EDM (Karras et al., 2022) and LDM (Rombach et al., 2022).

**Trajectory generation.** Following Zhou et al. (2024b), we use DPM-Solver++(3M) (Lu et al., 2025) as the teacher solver with $M = 4$ and $t_{min} = 0.006$. For text-to-image generation with Stable Diffusion (Rombach et al., 2022), we adopt DPM-Solver++(2M) (Lu et al., 2025) (the default Stable Diffusion setting) with $M = 3$ and $t_{min} = 0.01$. We generate 3K trajectories on MS-COCO (Lin et al., 2014), 7K on FFHQ (Karras et al., 2019), and 5K on other datasets. Additional trajectory generation details are provided in the Algorithm 2.

**Training settings.** We optimize all datasets using the AdamW optimizer (Loshchilov & Hutter, 2017). The learning rate is set to 1e-5 for Stable Diffusion (Rombach et al., 2022) and 5e-5 for other datasets. TrajP-L uses an L1 loss for parameter optimization across all datasets. All training is conducted on a single NVIDIA RTX 3090 GPU. Further training details can be found in Table 7.

**Evaluation.** For Stable Diffusion (Rombach et al., 2022), we evaluate TrajP-L over NFE $\in \{8, 12, 16, 20\}$. For other datasets, we evaluate over NFE $\in \{2, 3, 4, 5, 6\}$, using AFS (Dockhorn et al., 2022) to save one NFE. Sample quality is measured using the Fréchet Inception Distance (FID) (Heusel et al., 2017). For Stable Diffusion, FID is computed by generating 30K images from MS-COCO (Lin et al., 2014) validation prompts with a guidance scale of 7.5. For other datasets, we sample 50K images to calculate FID. The validation set serves as reference, and FID computation follows the protocols in Liu et al. (2023); Sauer et al. (2024).

## 4.2 MAIN RESULTS

In Table 1, we report FID scores of TrajP-L compared with several baseline solvers on the CIFAR10 32×32 (Krizhevsky & Hinton, 2009). The results demonstrate that our learned sampling direction consistently yields substantial improvements across all NFE settings. Notably, at NFE = 4 and NFE = 6, TrajP-L achieves FID scores of 3.54 and 3.09, respectively, compared to 25.00 and 7.28 for iPNDM (Liu et al., 2022), and 9.70 and 74.70 for EPD-Solver (Zhu et al., 2025). Moreover, under extremely low NFE conditions (e.g., NFE = 2), TrajP-L attains an FID of 5.02, substantially outperforming EPD-Solver (108.93). Additional comparison results are provided in Appendix D.

Table 1: Experimental results of FID↓ on CIFAR10 32×32 (Krizhevsky & Hinton, 2009).

| Method | NFE | | | | |
|---|---|---|---|---|---|
| | 2 | 3 | 4 | 5 | 6 |
| **Training-Free** | | | | | |
| DDIM (Song et al., 2020a) | 169.50 | 93.73 | 67.24 | 50.14 | 35.83 |
| EDM (Karras et al., 2022) | 468.41 | 306.08 | 319.75 | 98.18 | 100.32 |
| DPM-Solver-2 (Lu et al., 2022) | 290.34 | 155.35 | 146.04 | 57.67 | 60.33 |
| iPNDM (Liu et al., 2022) | 154.24 | 48.13 | 25.00 | 13.74 | 7.28 |
| DEIS (Zhang & Chen, 2022) | 143.52 | 56.46 | 25.83 | 14.46 | 9.51 |
| DPM-Solver++(3M) (Lu et al., 2025) | 147.33 | 110.58 | 46.73 | 25.14 | 12.22 |
| UniPC (Zhao et al., 2023) | 148.40 | 110.10 | 45.43 | 24.09 | 11.38 |
| **Training-Based** | | | | | |
| AMED-Solver (Zhou et al., 2024a) | 387.11 | 25.04 | 15.33 | 7.77 | 7.94 |
| EPD-Solver (Zhu et al., 2025) | 108.93 | 10.62 | 9.70 | 4.49 | 4.70 |
| **TrajP-L (ours)** | **5.02** | **3.91** | **3.54** | **3.34** | **3.09** |

We evaluate TrajP-L on Stable Diffusion v1.5 (Rombach et al., 2022) with a classifier-free guidance weight of 7.5, reporting FID scores on the MS-COCO validation set in Table 2. Furthermore, we compare the sample quality between DPM-Solver++(2M) (Lu et al., 2025) and AMED-Plugin (applied on DPM-Solver++(2M)) (Zhou et al., 2024a). The results consistently demonstrate the superiority of our proposed method.

Table 2: Sample quality measured by FID↓ on Stable Diffusion v1.5 with a guidance scale of 7.5. [†]We directly borrowed the results reported by Zhou et al. (2024a).

| Method | NFE (1 step = 2 NFE) | | | |
|---|---|---|---|---|
| | 8 | 12 | 16 | 20 |
| DPM-Solver++(2M) (Lu et al., 2025) | 21.59 | 15.85 | 14.81 | 14.28 |
| [†]AMED-Solver (Zhou et al., 2024a) | 18.92 | 14.84 | 13.96 | 13.24 |
| **TrajP-L (Ours)** | **17.91** | **14.14** | **12.97** | **12.39** |

## 4.3 ABLATION STUDIES

**Impact of trajectory counts:** Although the sampling trajectories of DMs exhibit highly similar spatial shapes (Chen et al., 2024), fine-tuning DMs with an extremely small number of trajectories remains challenging due to variations in their starting points and directions. To investigate the number of trajectories required to cover the complete sampling space, we conducted ablation studies on CIFAR10 32×32 (Krizhevsky & Hinton, 2009), ImageNet 64×64 (Deng et al., 2009), and LSUN-Bedroom (Yu et al., 2015). We then determined the optimal number of trajectories based on the FID scores (Heusel et al., 2017) obtained from training with different trajectory counts. The results are illustrated in Figure 5, with additional details provided in Table 16.

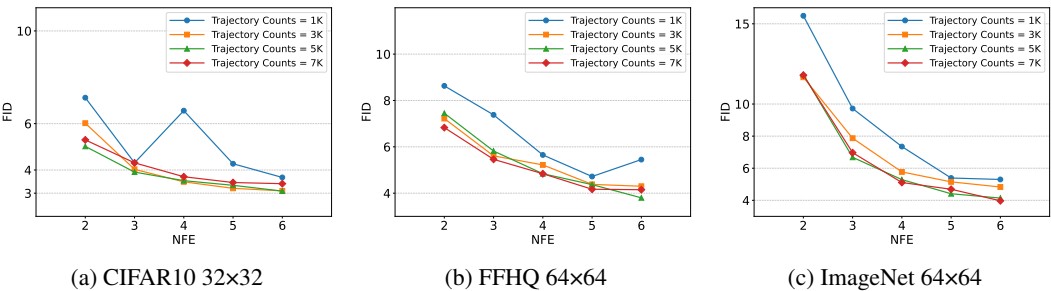

| (a) CIFAR10 32×32 | (b) FFHQ 64×64 | (c) ImageNet 64×64 |

Figure 5: Ablation study of trajectory counts on CIFAR10 32×32 (Krizhevsky & Hinton, 2009), FFHQ 64×64 (Karras et al., 2019), and ImageNet 64×64 (Deng et al., 2009) evaluated by FID↓.

**Impact of LoRA's rank and alpha:** Rank and alpha are key hyperparameters in LoRA that critically influence fine-tuning performance, affecting model accuracy, parameter efficiency, and generalization. Specifically, rank denotes the dimensionality of LoRA's low-rank matrices, while alpha serves as a scaling factor for the parameter updates. We identified the optimal rank–alpha combination via grid search across multiple datasets. Table 3 reports the results on ImageNet 64×64 (Deng et al., 2009), with additional findings provided in Appendix D.1.

Table 3: Ablation Study of LoRA's rank and alpha on ImageNet 64×64 (Deng et al., 2009), using 5K trajectories and evaluated by FID↓.

| Rank | Alpha | NFE | | | | |
|---|---|---|---|---|---|---|
| | | 2 | 3 | 4 | 5 | 6 |
| 8 | 4 | 13.46 | 8.82 | 5.50 | 6.79 | **3.87** |
| | 8 | 14.13 | 9.32 | 6.10 | 7.85 | 4.21 |
| 16 | 8 | 11.84 | 6.68 | **5.29** | **4.41** | 4.14 |
| | 16 | 12.42 | 8.18 | 5.62 | 4.84 | 4.45 |
| 24 | 12 | **11.54** | **6.35** | 5.31 | 4.68 | 4.47 |
| | 24 | 12.26 | 7.15 | 5.87 | 4.79 | 4.76 |

**Impact of LoRA and TrajP:** In Table 4, we report the performance of using TrajP, LoRA, and TrajP-L individually on the CIFAR10 32×32 (Krizhevsky & Hinton, 2009). The results demonstrate that TrajP-L consistently outperforms the individual components, achieving superior FID scores across all NFE settings.

**Impact of the number of principal components:** To investigate the sensitivity of TrajP to the number of trajectory principal components used, we present the ablation study results on the CIFAR10 32×32 Krizhevsky & Hinton (2009) in Table 5.

Table 4: Ablation study of TrajP-L on CIFAR10 32×32 (Krizhevsky & Hinton, 2009), using 5K trajectories and evaluated by FID↓.

| Used Modules | NFE | | | | |
|---|---|---|---|---|---|
| | 2 | 3 | 4 | 5 | 6 |
| w.o. TrajP-L (DDIM) | 169.50 | 93.73 | 67.24 | 50.14 | 35.83 |
| w.o. LoRA | 63.96 | 28.31 | 15.33 | 9.02 | 6.55 |
| w.o. TrajP | 5.26 | 4.14 | 3.64 | 3.52 | 3.19 |
| **TrajP-L** | **5.02** | **3.91** | **3.54** | **3.34** | **3.09** |

Table 5: Ablation study on the number of principal components on CIFAR10 32×32 (Krizhevsky & Hinton, 2009) with 5K trajectories based on FID↓.

| PC Nums. | NFE | | | | |
|---|---|---|---|---|---|
| | 2 | 3 | 4 | 5 | 6 |
| 2 | **4.88** | 4.16 | 3.50 | 3.38 | 3.19 |
| 3 | 5.02 | **3.91** | 3.54 | 3.34 | **3.09** |
| 4 | 4.94 | 4.08 | **3.41** | **3.13** | 3.17 |

### 4.4 QUALITATIVE ANALYSIS

In Figure 6, we present a comparison of images generated by DDIM (Song et al., 2020a), iPNDM (Liu et al., 2022), and our proposed TrajP-L using pre-trained models on CIFAR10 32×32 (Krizhevsky & Hinton, 2009), FFHQ 64×64 (Karras et al., 2019), and LSUN-Bedroom 256×256 (Yu et al., 2015). Under identical NFE settings, TrajP-L consistently produces images with superior visual quality, and this advantage becomes increasingly pronounced as NFE decreases. For example, at NFE = 2, TrajP-L generates clear, high-fidelity images, whereas the outputs from other samplers appear noticeably blurred. Additional visual comparisons are provided in Appendix D.2.

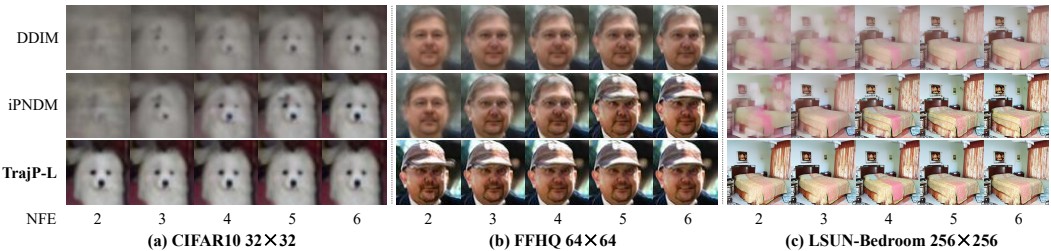

Figure 6: Comparison of generated samples among DDIM (Song et al., 2020a), iPNDM (Liu et al., 2022) and TrajP-L. Compared to other methods, TrajP-L achieves high-quality results even at NFE = 2.

## 5 CONCLUSION

In this paper, based on the observation that diffusion model sampling trajectories exhibit significant shape similarity and leveraging LoRA's fine-tuning mechanism, we propose TrajP-L, a trajectory-based plugin with LoRA. TrajP-L acquires prior knowledge of trajectory shapes by incorporating fitted expressions of teacher model sampling trajectories in 3D space, thereby enabling distillation with only a small number of teacher trajectories. This design effectively mitigates errors arising in few-step sampling with large intervals, leading to a substantial reduction in NFE required for diffusion model sampling. Extensive experimental results demonstrate that TrajP-L consistently delivers strong performance on both unconditional and conditional pre-trained diffusion models.

ETHICS STATEMENT

Like other generative models such as GANs (Goodfellow et al., 2020) and VAEs (Kingma & Welling, 2013), DMs may be used to generate malicious content and cause certain social harms. The TrajP-L proposed in this paper can significantly accelerate the generation process of DMs with a small amount of training resources, thereby reducing the cost of generating malicious or false data. However, with the development of deepfake detection technology, the potential negative social impacts of our technology can be mitigated to a certain extent.

REPRODUCIBILITY STATEMENT

Our code is based on the official implementations of AMED (Zhou et al., 2024a), SFD (Zhou et al., 2024b) and EPD (Zhu et al., 2025). We used the unconditional checkpoint of EDM (Karras et al., 2022) and the conditional checkpoint of LDM (Rombach et al., 2022). Detailed experimental settings and algorithm implementations are described in Appendix C, and the main code has been uploaded along with the supplementary material.

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

## A  RELATED WORKS

Compared to GANs (Goodfellow et al., 2020) and VAEs (Kingma & Welling, 2013), DMs generate images of superior quality (Dhariwal & Nichol, 2021), but require hundreds of denoising steps (Ho et al., 2020; Song & Ermon, 2019). Consequently, numerous studies have sought to accelerate DMs sampling, with approaches falling broadly into solver-based and distillation-based methods.

**Solver-based methods.** Song et al. (2020b) unified DMs' noise injection and denoising processes into a stochastic differential equation formulation (see Eqs. (2–5)), and showed the sampling procedure can be interpreted as solving a reverse PF-ODE (see Eqs. (6–7)). This perspective inspired a range of solver-based methods. Among them, Euler method (Song et al., 2020a), Heun method (Karras et al., 2022), iPNDM (Liu et al., 2022), DEIS (Zhang & Chen, 2022), UniPC (Zhao et al., 2023), and Taylor expansion-based solvers (e.g., DPM-Solver (Lu et al., 2022), DPM-Solver++ (Lu et al., 2025)) are widely adopted for their training-free nature. Furthermore, (Wang et al., 2024a) introduced an orthogonal dimension to these PF-ODE solvers by incorporating past and future scores.

However, due to discretization errors, training-free approaches still suffer from suboptimal image quality at low NFE. To address this, many studies have explored improved time discretization schedules for faster sampling. For instance, GITS (Chen et al., 2024) leverages the similarity of denoising trajectories in DMs and uses dynamic programming to optimize time allocation; AYS (Sabour et al., 2024) is the first to apply stochastic calculus to time scheduling, overcoming the limitation of traditional heuristic-based strategies. Additionally, other works reduce discretization errors by learning combinations of past or future denoising directions. For example, AMED-Solver (Zhou et al., 2024a) uses the mean value theorem to train a lightweight network that predicts the optimal intermediate future timestamp between current and next denoising steps, taking its denoising direction as the current update; EPD-Solver (Zhu et al., 2025) computes K intermediate future timestamps in parallel and samples from a weighted combination of these K future directions and the current one; S4S (Frankel et al., 2025), by contrast, focuses on "current and past steps", reducing discretization errors via learning an optimized weighted combination of their denoising directions.

**Distillation-based methods.** Both training-free and training-based solver methods suffer from rapidly escalating discretization errors at low NFE, severely degrading image quality. Distillation-based methods, by contrast, address this by retraining or fine-tuning DMs to recalibrate denoising directions, ensuring closer alignment with ground-truth transitions between denoising steps. These methods are generally divided into three categories: trajectory distillation, consistency distillation, and distribution matching.

Trajectory distillation (Luhman & Luhman, 2021) achieves acceleration by guiding the student model's one-step denoising direction to approximate the teacher model's multi-step trajectory. Existing works follow two main strategies: (i) multi-stage distillation, which progressively decreases the number of sampling steps (Salimans & Ho, 2022; Meng et al., 2023; Li et al., 2023); and (ii) trajectory imitation, where the student is trained to replicate the teacher's complete sampling path (Zhou et al., 2024b; Wang et al., 2024b).

Consistency distillation accelerates sampling by enforcing alignment of denoising directions across all time steps (Song et al., 2023), demonstrating strong efficiency gains. Luo et al. (2023a) extended this idea to the latent space and validated its effectiveness, while Kim et al. (2023) generalized consistency distillation to arbitrary timestamps by modeling both fine-grained and long-range jumps along the PF-ODE trajectory. Despite its ability to enable single-step generation, consistency distillation demands substantial training resources, often exceeding the cost of training the original teacher model (Song et al., 2023; Zhou et al., 2024b). In addition, LCM-LoRA (Luo et al., 2023b; Thakur & Vashisth, 2024) integrates LoRA into latent space consistency distillation, which effectively accelerates the process and reduces the distillation cost.

Distribution-matching methods (Sauer et al., 2024; Yin et al., 2024b;a) align generated and real samples at the distribution level, enabling high-quality one-step generation but often requiring auxiliary loss functions. For example, DMD (Yin et al., 2024b) minimizes an approximate KL divergence to better match the generated and real distributions. However, such approaches are susceptible to mode collapse and, like other distillation-based methods, incur substantial computational costs.

While TrajP-L belongs to the family of trajectory distillation methods, it departs from existing approaches in two key ways: (i) it introduces LoRA (Hu et al., 2022) into trajectory distillation, pre-

serving the parameter space of the teacher model, (ii) it incorporates a trajectory-shape prior based on the inherent similarity of denoising paths in diffusion models. This design implicitly applies PCA (Abdi & Williams, 2010) to diffusion trajectories, which—when combined with a trajectory fitting function—guides corrections to the denoising direction. As a result, TrajP-L attains comparable performance with only 3K–7K teacher trajectories (see Table 7), whereas conventional trajectory distillation typically requires around 200K (Zhou et al., 2024b).

# B  ALGORITHMS OF TRAIP-L

All the algorithms involved in the main text are illustrated in the following.

---

**Algorithm 2** Trajectory Generation

---

**Require:** pre-trained model $D_\theta$, number of student sampling steps $N$, number of teacher sampling steps between every two noise levels of student $M$, number of teacher sampling steps $N' = N + M \times (N - 1)$, noise schedule $\{t_n\}_{n=0}^{N'}$.

1: Init: $\mathbb{L} = \{\}$          ▷ Initialize the teacher's sampling trajectory list
2: **repeat**
3:      Sample: $x_{t_{N'}} = \tilde{x}_{t_{N'}} \sim \mathcal{N}(0; t_{N'}^2 \mathbf{I})$
4:      Init: $count = 0$, $L = \{x_{t_{N'}}\}$
5:      **for** $n = N'$ to 1 **do**
6:          $\hat{x}_{t_0} = D_\theta(x_{t_n}, t_n)$
7:          $x_{t_{n-1}} = x_{t_n} + (t_n - t_{n-1})(x_{t_n} - \hat{x}_{t_0})/t_n$
8:          $x_{t_n} = x_{t_{n-1}}$
9:          $count = count + 1$
10:          **if** $count == M + 1$ **then**     ▷ Save the trajectory of the teacher model every $M$ steps
11:             $L.\text{append}(x_{t_n})$
12:             $count = 0$
13:          **end if**
14:      **end for**
15:      $\mathbb{L}.\text{append}(L)$
16: **until** convergence
17: **return** $\mathbb{L}$

---

**Algorithm 3** Training

---

**Require:** pre-trained model with LoRA $L_\theta$, TrajP plugin $T_\theta$, trajectory fitting model $C_\theta$, teacher's sampling trajectory list $\mathbb{L}$, number of student sampling steps $N$, noise schedule $\{t_n\}_{n=0}^{N}$.

1: **repeat**
2:      **for** $i = 0$ to $len(\mathbb{L})$ **do**
3:          $L = \mathbb{L}[i]$, $x_{t_N} = L[0]$, $Q = \{x_{t_N}\}$
4:          **for** $n = N$ to 1 **do**
5:             $\hat{x}_{t_0} = L_\theta(x_{t_n}, t_n)$
6:             $\hat{d}_{t_n} = (x_{t_n} - \hat{x}_{t_0})/t_n$      ▷ The denoising direction calculated using solely $L_\theta$
7:             $u, v, w = \text{detach}(C_\theta(t_n, t_{n-1}))$ ▷ Obtain the trajectory information from $t_n$ to $t_{n-1}$
8:             $\tilde{d}_{t_n} = T_\theta(\hat{d}_{t_n}, Q, u, v, w)$          ▷ Use $T_\theta$ to correct the denoising direction
9:             $Q.\text{append}(\tilde{d}_{t_n})$                ▷ Cache the denoising direction
10:             $\tilde{x}_{t_{n-1}} = x_{t_n} + (t_n - t_{n-1})\tilde{d}_{t_n}$
11:             $x_{t_{n-1}} = L[N - n + 1]$
12:             $\theta = \theta - \alpha \nabla_\theta d(\tilde{x}_{t_{n-1}}, x_{t_{n-1}})$ ▷ Synchronously update the parameters of $L_\theta$ and $T_\theta$
13:             $x_{t_{n-1}} = \tilde{x}_{t_{n-1}}$
14:          **end for**
15:      **end for**
16: **until** convergence

---

---

**Algorithm 4** Sampling

---

**Require:** pre-trained model with LoRA $L_\theta$, TrajP plugin $T_\theta$, trajectory fitting model $C_\theta$, number of student sampling steps $N$, noise schedule $\{t_n\}_{n=0}^N$.
1: Sample: $x_{t_N} = \tilde{x}_{t_N} \sim \mathcal{N}(0; t_N^2 \mathbf{I})$
2: Init: $Q = \{x_{t_N}\}$
3: **for** $n = N$ to 1 **do**
4:    $\hat{x}_{t_0} = L_\theta(x_{t_n}, t_n)$
5:    $\hat{d}_{t_n} = (x_{t_n} - \hat{x}_{t_0})/t_n$
6:    $u, v, w = \text{detach}(C_\theta(t_n, t_{n-1}))$
7:    $d_{t_n} = T_\theta(\hat{d}_{t_n}, Q, u, v, w)$
8:    $Q.\text{append}(d_{t_n})$
9:    $x_{t_{n-1}} = x_{t_n} + (t_n - t_{n-1})d_{t_n}$
10: **end for**
11: **return** $x_{t_0}$

---

## C   ADDITIONAL EXPERIMENT DETAILS

### C.1   LICENSE

In this section, we list the links and licenses for the datasets and pre-trained checkpoints used in the main text in Table 6.

Table 6: The used datasets, checkpoints, as well as their links and licenses.

| Name | URL | License |
|---|---|---|
| CIFAR10 32x32  (Krizhevsky & Hinton, 2009) | cs.toronto.edu | \ |
| FFHQ 64x64 (Karras et al., 2019) | github.com/NVlabs | CC BY-NC-SA 4.0 |
| ImageNet 64x64 (Deng et al., 2009) | image-net.org | \ |
| LSUN-Bedroom 256x256 (Yu et al., 2015) | yf.io | \ |
| MS-COCO 512x512 (Lin et al., 2014) | cocodataset.org | CC BY 4.0 |
| edm-cifar10-32x32-uncond-vp.pkl (Karras et al., 2022) | github.com/NVlabs | CC BY-NC-SA 4.0 |
| edm-ffhq-64x64-uncond-vp.pkl (Karras et al., 2022) | github.com/NVlabs | CC BY-NC-SA 4.0 |
| edm-imagenet-64x64-cond-adm.pkl (Karras et al., 2022) | github.com/NVlabs | CC BY-NC-SA 4.0 |
| lsun_bedrooms.zip (Rombach et al., 2022) | ommer-lab.com | \ |
| stable-diffusion-v1-5 (Rombach et al., 2022) | huggingface.co | \ |
| vq-f4 (Rombach et al., 2022) | ommer-lab.com | \ |

### C.2   SETTINGS

In Table 7, we summarize all the hyperparameters used in our main experiments (Section 4). In Table 8, we report the parameter sizes of the pre-trained model, LoRA, and the TrajP plug-in across CIFAR-10 32×32 (Krizhevsky & Hinton, 2009), FFHQ 64×64 (Karras et al., 2019), ImageNet 64×64 (Deng et al., 2009), LSUN-Bedroom 256×256 (Yu et al., 2015), and Stable Diffusion v1.5 (Rombach et al., 2022). The parameter counts of LoRA are computed based on the rank and alpha values provided in Table 7.

### C.3   TRAJECTORY SIMILARITY

We randomly sample $N$ trajectories from those generated by Algorithm 2. To standardize their initial positions, we subtract the sampling origin so that all trajectories share a common starting point. The resulting set of trajectories is denoted as $X_0, X_1, \ldots, X_N$, where each $X_i \in \mathbb{R}^{T \times D}$ represents a trajectory of $T$ time steps in a $D$-dimensional space. Taking $X_0$ as the reference, we align each $X_i$ ($i \geq 1$) to $X_0$ via a rigid rotation $R_i$ in $D$ dimensions:

Table 7: Experiment settings used in the main text. $^*$: We force a batch size of 128 by accumulating the gradient for 32 rounds.

| Hyperparameter | CIFAR10 | FFHQ | ImageNet64 | LSUN-Bedroom | Stable Diffusion |
|---|---|---|---|---|---|
| Trajectory count | 5K | 7K | 5K | 5K | 3K |
| Teahcer solver | DPM-Solver++(3M) | DPM-Solver++(3M) | DPM-Solver++(3M) | DPM-Solver++(3M) | DPM-Solver++(2M) |
| Learning rate | 5e-5 | 5e-5 | 5e-5 | 5e-5 | 1e-5 |
| Optimizer | AdamW | AdamW | AdamW | AdamW | AdamW |
| Loss metric | L1 | L1 | L1 | L1 | L1 |
| Rank | 16 | 16 | 16 | 32 | 64 |
| Alpha | 16 | 8 | 8 | 32 | 64 |
| Batch size | 128 | 128 | 32 | 8 | 4* |
| Number of GPUs | 1 | 1 | 1 | 1 | 1 |

Table 8: The sizes of pre-trained model, LoRA, and TrajP.

| Model | CIFAR10 | FFHQ | ImageNet64 | LSUN-Bedroom | Stable Diffusion |
|---|---|---|---|---|---|
| Pre-trained model | 230.4MB | 256.6MB | 1,236.3MB | 1,304.8MB | 4,173.7 MB |
| LoRA | 7.17MB | 8.65MB | 18.94MB | 54.31MB | 112.99MB |
| TrajP | 0.43MB | 0.43MB | 0.43MB | 0.43MB | 0.44MB |

$$R_i = \underset{R_i \in \mathbb{SO}(D)}{\arg\min} \|R_i X_i^\top - X_0^\top\|_2^2,$$
$$X_i' = (R_i X_i^\top)^\top, \tag{20}$$

where $X_i' \in \mathbb{R}^{T \times D}$ denotes the aligned trajectory. Next, we perform PCA-based dimensionality reduction on $X_0$. First, we center the reference trajectory:

$$\tilde{X}_0 = X_0 - \mathbf{1}_T \bar{X}_0, \tag{21}$$

where $\bar{X}_0$ is the mean vector over time steps. The covariance matrix is then computed as:

$$\Sigma_{\tilde{X}_0} = \frac{1}{T-1} \tilde{X}_0^\top \tilde{X}_0. \tag{22}$$

Eigenvalue decomposition of $\Sigma_{\tilde{X}_0}$ gives:

$$\Sigma_{\tilde{X}_0} P = P\Lambda, \tag{23}$$

where $P = [PC_1, PC_2, \ldots, PC_D]$ contains the principal components and $\Lambda = \mathrm{diag}(\lambda_1, \lambda_2, \ldots, \lambda_D)$ is the diagonal eigenvalue matrix. Taking the top three principal components $P = [PC_1, PC_2, PC_3]$, we project the centered reference trajectory as:

$$Y_0 = [U_0, V_0, W_0] = \tilde{X}_0 P. \tag{24}$$

Similarly, the projections of $X_i$ and $X_i'$ onto the same principal components are:

$$Y_i = [U_i, V_i, W_i] = X_i P,$$
$$Y_i' = [U_i', V_i', W_i'] = X_i' P. \tag{25}$$

We visualize $Y_i$ and $Y_i'$ in $\mathbb{R}^3$ (Figure 7) and compute L1 distances at corresponding time steps (Table 9). The results confirm that, after alignment and projection onto the same principal components, all trajectories preserve similar reduced-dimensional coordinates. This demonstrates that the diffusion trajectories maintain structural similarity both in the original high-dimensional space and in the reduced three-dimensional space.

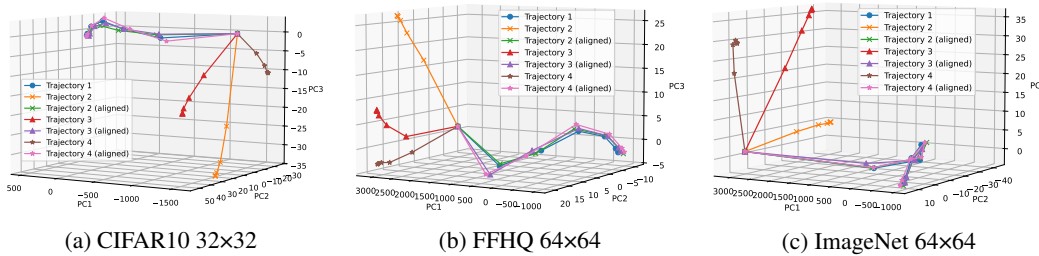

(a) CIFAR10 32×32        (b) FFHQ 64×64        (c) ImageNet 64×64

Figure 7: Set $N = 4$, and perform trajectory visualization before and after alignment on CIFAR10 32×32 (Krizhevsky & Hinton, 2009), FFHQ 64×64 (Karras et al., 2019), and ImageNet 64×64 (Deng et al., 2009).

Table 9: The L1 distance between the trajectories before and after alignment and the target alignment trajectory.

|  | CIFAR10 32×32 | FFHQ 64×64 | ImageNet 64×64 |
|---|---|---|---|
| Origin distance | 95.29 | 94.79 | 94.74 |
| Aligned distance | 0.82 | 0.32 | 0.32 |

## C.4 TRAJECTORY FITTING

We report the parameter values for the initialization and iterative optimization of the trajectory fitting function in Tables 10-11, respectively. Among them, the initialization parameters are obtained by fitting one sampling trajectory in 3D space, and the parameters for iterative optimization are obtained by minimizing the $L_2$ loss between the trajectory fitting function and 64 sampling trajectories in 3D space. The real sampling trajectories and the trajectories of the fitting function are visualized in Figures 8-9.

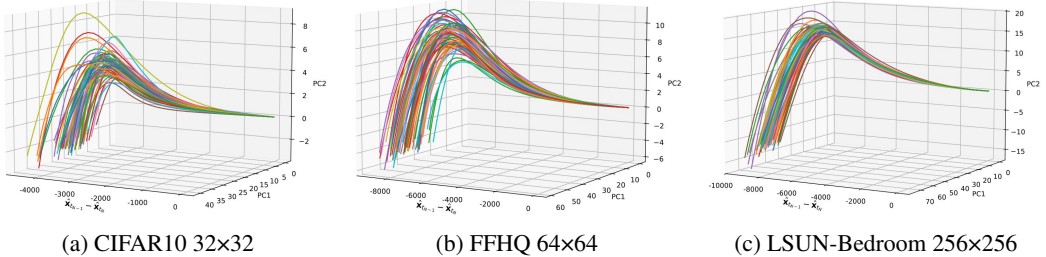

(a) CIFAR10 32×32        (b) FFHQ 64×64        (c) LSUN-Bedroom 256×256

Figure 8: The 3D sampling trajectories on CIFAR10 32×32 (Krizhevsky & Hinton, 2009), FFHQ 32×32 (Karras et al., 2019), and LSUN-Bedroom 256×256 (Yu et al., 2015).

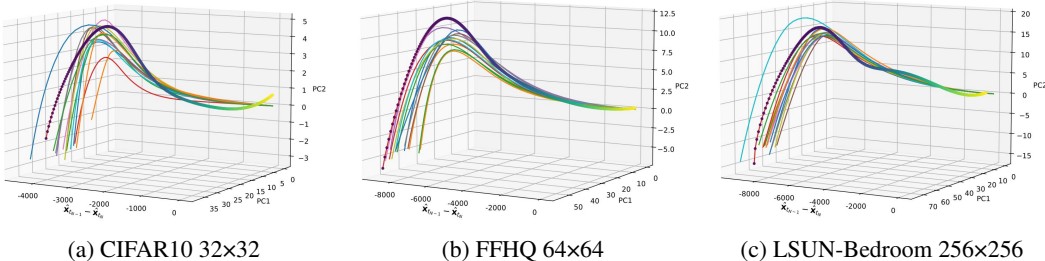

(a) CIFAR10 32×32        (b) FFHQ 64×64        (c) LSUN-Bedroom 256×256

Figure 9: The fitted 3D sampling trajectories (bolded with color gradients) on CIFAR10 32×32 (Krizhevsky & Hinton, 2009), FFHQ 32×32 (Karras et al., 2019), and LSUN-Bedroom 256×256 (Yu et al., 2015).

Table 10: Initial values of trajectory parameters for different datasets (4 decimal places retained).

| Dataset | $P$ (Quadratic Coefficients) | | | $Q$ (Linear Coefficients) | | |
|---|---|---|---|---|---|---|
| | Dim 1 | Dim 2 | Dim 3 | Dim 1 | Dim 2 | Dim 3 |
| CIFAR10 32×32 (Krizhevsky & Hinton, 2009) | -0.0070 | 0.0016 | -0.0493 | -59.8158 | -0.2034 | -58.7852 |
| FFHQ 64×64 (Karras et al., 2019) | -0.0524 | 0.0012 | 0.0017 | -137.6131 | -0.1625 | -0.2120 |
| ImageNet 64×64 (Deng et al., 2009) | -0.0518 | 0.0020 | 0.0026 | -137.6131 | -0.2742 | -0.3307 |
| LSUN-Bedroom 256×256 (Yu et al., 2015) | -0.6811 | 0.0022 | -0.3466 | -810.3361 | -0.3022 | -409.4982 |
| MS-COCO 512×512 (Lin et al., 2014) | -459.1779 | -17.7515 | -2.7182 | 42514.7574 | -1905.9572 | -4.7869 |

| Dataset | $R$ (Constant Terms) | | | $A$ (Exponential Coefficients) | | |
|---|---|---|---|---|---|---|
| | Dim 1 | Dim 2 | Dim 3 | Dim 1 | Dim 2 | Dim 3 |
| CIFAR10 32×32 (Krizhevsky & Hinton, 2009) | 2811.5302 | 6.3434 | -38092.3923 | 1642.0989 | 23.2146 | 38090.8012 |
| FFHQ 64×64 (Karras et al., 2019) | 184.0770 | 5.5703 | 6.6254 | 8680.5633 | 49.7301 | -12.7075 |
| ImageNet 64×64 (Deng et al., 2009) | 381.9386 | 9.3447 | 10.4323 | 8482.7340 | 49.1384 | -16.5279 |
| LSUN-Bedroom 256×256 (Yu et al., 2015) | -397728.4850 | 10.3464 | -264142.0770 | 406917.7000 | 55.3313 | 264126.9000 |
| MS-COCO 512×512 (Lin et al., 2014) | -434322.0130 | -118349.6330 | -163.9738 | 436148.1070 | 118374.6260 | 165.0992 |

| Dataset | $\lambda$ (Exponential Parameters) | | | $G_f$ (Gamma Coefficients) | | |
|---|---|---|---|---|---|---|
| | Dim 1 | Dim 2 | Dim 3 | Dim 1 | Dim 2 | Dim 3 |
| CIFAR10 32×32 (Krizhevsky & Hinton, 2009) | 0.0026 | -0.2468 | 0.0015 | 1.7123 | 1.3418 | 3.0624 |
| FFHQ 64×64 (Karras et al., 2019) | 0.0031 | -0.1699 | -0.2886 | 7.9330 | -4.5161 | 9.9211 |
| ImageNet 64×64 (Deng et al., 2009) | 0.0032 | -0.1596 | -0.2983 | 7.7655 | -4.6567 | 9.2557 |
| LSUN-Bedroom 256×256 (Yu et al., 2015) | 0.0017 | -0.2404 | 0.0016 | 87.2055 | -5.6255 | 19.9333 |
| MS-COCO 512×512 (Lin et al., 2014) | 0.0227 | 0.0162 | 0.1093 | -52503.2025 | 65.9068 | 32.8175 |

| Dataset | $G_g$ (Gamma Exponents) | | | $G_h$ (Gamma Divisors) | | |
|---|---|---|---|---|---|---|
| | Dim 1 | Dim 2 | Dim 3 | Dim 1 | Dim 2 | Dim 3 |
| CIFAR10 32×32 (Krizhevsky & Hinton, 2009) | 0.3657 | 1.6640 | 0.8402 | 20.4928 | 0.5291 | 6.4688 |
| FFHQ 64×64 (Karras et al., 2019) | 0.3023 | 1.0018 | 0.8709 | 12.8904 | 2.9322 | 3.2643 |
| ImageNet 64×64 (Deng et al., 2009) | 0.3054 | 0.8771 | 0.8231 | 12.6909 | 3.4160 | 3.0917 |
| LSUN-Bedroom 256×256 (Yu et al., 2015) | 0.1979 | 1.9883 | 0.6881 | 13.8540 | 1.2268 | 5.5515 |
| MS-COCO 512×512 (Lin et al., 2014) | 1.0005 | 0.0085 | 1.8809 | 148.6493 | 2.0657 | 0.6105 |

Table 11: Learned values of trajectory parameters for different datasets (4 decimal places retained).

| Dataset | $P$ (Quadratic Coefficients) | | | $Q$ (Linear Coefficients) | | |
|---|---|---|---|---|---|---|
| | Dim 1 | Dim 2 | Dim 3 | Dim 1 | Dim 2 | Dim 3 |
| CIFAR10 32×32 (Krizhevsky & Hinton, 2009) | -0.0070 | 0.0016 | -0.0493 | -59.8158 | -0.2034 | -58.7852 |
| FFHQ 64×64 (Karras et al., 2019) | -0.0471 | 0.0013 | 0.0018 | -137.6131 | -0.1685 | -0.2197 |
| ImageNet 64×64 (Deng et al., 2009) | -0.0474 | 0.0021 | 0.0026 | -137.6131 | -0.2814 | -0.3385 |
| LSUN-Bedroom 256×256 (Yu et al., 2015) | -0.6773 | 0.0023 | -0.3468 | -810.3361 | -0.3098 | -409.4982 |
| MS-COCO 512×512 (Lin et al., 2014) | -459.1779 | -17.6980 | -2.7238 | 42514.7580 | -1905.9572 | -4.8550 |

| Dataset | $R$ (Constant Terms) | | | $A$ (Exponential Coefficients) | | |
|---|---|---|---|---|---|---|
| | Dim 1 | Dim 2 | Dim 3 | Dim 1 | Dim 2 | Dim 3 |
| CIFAR10 32×32 (Krizhevsky & Hinton, 2009) | 2811.5303 | 6.3434 | -38092.3910 | 1642.0989 | 23.2146 | 38090.8010 |
| FFHQ 64×64 (Karras et al., 2019) | 184.0770 | 5.5626 | 6.6178 | 8680.5630 | 49.7302 | -12.6922 |
| ImageNet 64×64 (Deng et al., 2009) | 381.9386 | 9.3294 | 10.4238 | 8482.7340 | 49.1384 | -16.5279 |
| LSUN-Bedroom 256×256 (Yu et al., 2015) | -397728.5000 | 10.3312 | -264142.0600 | 406917.6900 | 55.3313 | 264126.9100 |
| MS-COCO 512×512 (Lin et al., 2014) | -434322.0000 | -118349.6300 | -163.9738 | 436148.0940 | 118374.6250 | 165.0992 |

| Dataset | $\lambda$ (Exponential Parameters) | | | $G_f$ (Gamma Coefficients) | | |
|---|---|---|---|---|---|---|
| | Dim 1 | Dim 2 | Dim 3 | Dim 1 | Dim 2 | Dim 3 |
| CIFAR10 32×32 (Krizhevsky & Hinton, 2009) | 0.0025 | -0.2467 | 0.0015 | 1.7123 | 1.3418 | 3.0624 |
| FFHQ 64×64 (Karras et al., 2019) | 0.0031 | -0.1777 | -0.2805 | 7.9254 | -4.5238 | 9.9058 |
| ImageNet 64×64 (Deng et al., 2009) | 0.0032 | -0.1674 | -0.2903 | 7.7579 | -4.6643 | 9.2405 |
| LSUN-Bedroom 256×256 (Yu et al., 2015) | 0.0017 | -0.2483 | 0.0016 | 87.2055 | -5.6331 | 19.9333 |
| MS-COCO 512×512 (Lin et al., 2014) | 0.0227 | 0.0162 | 0.1092 | -52503.2030 | 65.9095 | 32.7336 |

| Dataset | $G_g$ (Gamma Exponents) | | | $G_h$ (Gamma Divisors) | | |
|---|---|---|---|---|---|---|
| | Dim 1 | Dim 2 | Dim 3 | Dim 1 | Dim 2 | Dim 3 |
| CIFAR10 32×32 (Krizhevsky & Hinton, 2009) | 0.3657 | 1.6640 | 0.8402 | 20.4928 | 0.5291 | 6.4688 |
| FFHQ 64×64 (Karras et al., 2019) | 0.2964 | 1.0014 | 0.8633 | 12.8904 | 2.9398 | 3.2567 |
| ImageNet 64×64 (Deng et al., 2009) | 0.3020 | 0.8834 | 0.8155 | 12.6910 | 3.4236 | 3.0841 |
| LSUN-Bedroom 256×256 (Yu et al., 2015) | 0.1941 | 1.9959 | 0.6913 | 13.8540 | 1.2345 | 5.5522 |
| MS-COCO 512×512 (Lin et al., 2014) | 1.0004 | -0.0341 | 1.9873 | 148.6493 | 2.0799 | 0.5643 |

# D  ADDITIONAL EXPERIMENT RESULTS

We report the FID results of TrajP-L under different NFE on FFHQ 64×64 (Karras et al., 2019), ImageNet 64×64 (Deng et al., 2009), and LSUN-Bedroom 256×256 (Yu et al., 2015) in Tables 12-14. For CIFAR10 32×32 (Krizhevsky & Hinton, 2009), the experimental results are shown in Table 1.

Table 12: Experimental results of FID↓ on FFHQ 64×64 (Karras et al., 2019).

| Method | NFE | | | | |
|---|---|---|---|---|---|
| | 2 | 3 | 4 | 5 | 6 |
| **Training-Free** | | | | | |
| DDIM (Song et al., 2020a) | 104.99 | 78.03 | 57.44 | 43.85 | 29.42 |
| EDM (Karras et al., 2022) | 466.92 | 356.27 | 344.60 | 116.78 | 142.43 |
| DPM-Solver-2 (Lu et al., 2022) | 337.31 | 266.11 | 237.98 | 87.38 | 82.91 |
| iPNDM (Liu et al., 2022) | 102.77 | 45.96 | 28.36 | 17.14 | 10.03 |
| DEIS (Zhang & Chen, 2022) | 105.06 | 54.55 | 28.36 | 17.46 | 12.33 |
| DPM-Solver++(3M) (Lu et al., 2025) | 112.86 | 86.20 | 45.81 | 22.54 | 13.80 |
| UniPC (Zhao et al., 2023) | 114.46 | 86.17 | 44.68 | 21.44 | 12.93 |
| **Training-Based** | | | | | |
| AMED-Solver (Zhou et al., 2024a) | 379.29 | 48.63 | 25.83 | 15.41 | 10.83 |
| EPD-Solver (Zhu et al., 2025) | 323.89 | 21.72 | 14.54 | 7.76 | 6.49 |
| **TrajP-L (ours)** | **6.83** | **5.46** | **4.84** | **4.17** | **4.15** |

Table 13: Experimental results of FID↓ on ImageNet 64×64 (Deng et al., 2009).

| Method | NFE | | | | |
|---|---|---|---|---|---|
| | 2 | 3 | 4 | 5 | 6 |
| **Training-Free** | | | | | |
| DDIM (Song et al., 2020a) | 113.87 | 82.90 | 58.28 | 43.48 | 33.84 |
| EDM (Karras et al., 2022) | 438.73 | 246.90 | 249.4 | 91.66 | 89.49 |
| DPM-Solver-2 (Lu et al., 2022) | 247.16 | 140.18 | 129.28 | 42.12 | 43.19 |
| iPNDM (Liu et al., 2022) | 110.20 | 58.29 | 33.47 | 18.82 | 12.77 |
| DEIS (Zhang & Chen, 2022) | 110.22 | 44.59 | 23.43 | 14.65 | 10.46 |
| DPM-Solver++(3M) (Lu et al., 2025) | 143.31 | 91.25 | 56.08 | 25.24 | 14.92 |
| UniPC (Zhao et al., 2023) | 146.43 | 91.11 | 55.34 | 24.12 | 14.16 |
| **Training-Based** | | | | | |
| AMED-Solver (Zhou et al., 2024a) | 219.06 | 39.66 | 37.60 | 11.55 | 11.67 |
| EPD-Solver (Zhu et al., 2025) | 130.97 | 18.11 | 18.76 | 6.34 | 7.76 |
| **TrajP-L (ours)** | **11.84** | **6.68** | **5.29** | **4.41** | **4.14** |

## D.1 ADDITIONAL ABLATION STUDY RESULTS

Table 15 presents the LoRA's rank and alpha search results on CIFAR10 32×32 (Krizhevsky & Hinton, 2009) with 5K trajectories. The bold values in Table 15 represent the optimal FID results under each NFE: For instance, the minimum FID at NFE=2 is 4.62 (Rank=24, Alpha=36). Notably, Table 15 shows that the optimal FID results under different NFE are relatively scattered—these optimal results are not concentrated in a single fixed combination of LoRA rank and alpha parameters, but instead correspond to different parameter pairings. This phenomenon indicates that our method exhibits good robustness to the rank and alpha parameters of LoRA, meaning it can still achieve excellent performance under different NFE conditions without strictly relying on specific parameter combinations.

Table 16 presents the ablation experiments of TrajP-L on CIFAR10 32×32 (Krizhevsky & Hinton, 2009), FFHQ 64×64 (Karras et al., 2019), ImageNet 64×64 (Deng et al., 2009), and LSUN-Bedroom 256×256 (Yu et al., 2015) with different numbers of trajectories (1K, 3K, 5K, 7K). The results show that increasing the number of trajectories can improve the performance of TrajP-L to a certain extent; however, after the number of trajectories reaches a threshold, performance degradation may occur under some NFE. For example, on CIFAR10 32×32 (Krizhevsky & Hinton, 2009), when the number of trajectories increases from 3K to 5K, the FID results for NFE=2 and NFE=3 both decrease; when the number of trajectories further increases from 5K to 7K, the FID results under all NFE decrease.

Table 14: Experimental results of FID↓ on LSUN-Bedroom 256×256 (Yu et al., 2015).

| Method | NFE | | | | |
|---|---|---|---|---|---|
| | 2 | 3 | 4 | 5 | 6 |
| **Training-Free** | | | | | |
| DDIM (Song et al., 2020a) | 233.83 | 130.59 | 70.51 | 40.92 | 25.94 |
| EDM (Karras et al., 2022) | 371.62 | 266.87 | 267.33 | 157.42 | 150.22 |
| DPM-Solver-2 (Lu et al., 2022) | 267.63 | 198.34 | 177.32 | 68.42 | 61.65 |
| iPNDM (Liu et al., 2022) | 180.52 | 41.99 | 10.57 | 6.70 | 5.31 |
| DEIS (Zhang & Chen, 2022) | 190.51 | 48.38 | 10.98 | 7.29 | 6.32 |
| DPM-Solver++(3M) (Lu et al., 2025) | 175.18 | 115.65 | 48.49 | 20.28 | 9.94 |
| UniPC (Zhao et al., 2023) | 254.62 | 291.88 | 122.30 | 27.83 | 7.97 |
| **Training-Based** | | | | | |
| AMED-Solver (Zhou et al., 2024a) | 291.73 | 27.43 | 29.72 | 10.76 | 14.56 |
| EPD-Solver (Zhu et al., 2025) | 162.57 | 13.23 | 9.46 | 7.56 | 6.21 |
| **TrajP-L (ours)** | **9.88** | **6.63** | **5.27** | **4.52** | **4.42** |

Table 15: Ablation study of LoRA's rank and alpha on CIFAR10 32×32 (Krizhevsky & Hinton, 2009) with 5K trajectories based on FID↓.

| Rank | Alpha | NFE | | | | |
|---|---|---|---|---|---|---|
| | | 2 | 3 | 4 | 5 | 6 |
| | 4 | 6.22 | 4.59 | 3.78 | 4.14 | 3.27 |
| 8 | 8 | 5.11 | 4.22 | 3.77 | 5.02 | **3.04** |
| | 12 | 5.19 | 3.97 | 3.91 | 3.55 | 3.21 |
| | 8 | 5.18 | 5.10 | 3.85 | 3.62 | 3.27 |
| 16 | 16 | 5.02 | **3.91** | 3.54 | 3.34 | 3.09 |
| | 24 | 5.44 | 4.34 | **3.43** | 3.31 | 3.05 |
| | 12 | 5.07 | 4.30 | 4.02 | 3.40 | 3.29 |
| 24 | 24 | 4.84 | 4.18 | 3.65 | 3.42 | 3.10 |
| | 36 | **4.62** | 4.21 | 3.46 | **3.25** | 3.17 |

To further evaluate the fidelity and diversity of TrajP-L, we calculated precision, recall, density, and coverage under different NFE on the CIFAR10 32×32 (Krizhevsky & Hinton, 2009), following the guidelines in Naeem et al. (2020); results are presented in Table 17. Compared with solver-based methods, TrajP-L achieves significant acceleration in image generation while still maintaining comparable fidelity and diversity.

Table 16: Ablation study of TrajP-L on CIFAR10 32×32 (Krizhevsky & Hinton, 2009), FFHQ 64×64 (Karras et al., 2019), ImageNet 64×64 (Deng et al., 2009) and LSUN-Bedroom 256×256 (Yu et al., 2015) across different trajectory counts based on FID↓.

| Dataset | Traj. Count | NFE | | | | |
|---|---|---|---|---|---|---|
| | | 2 | 3 | 4 | 5 | 6 |
| CIFAR10 32×32 (Krizhevsky & Hinton, 2009) | 1K | 7.12 | 4.31 | 6.56 | 4.27 | 3.68 |
| | 3K | 6.02 | 4.03 | **3.49** | **3.21** | **3.09** |
| | 5K | **5.02** | **3.91** | 3.54 | 3.34 | **3.09** |
| | 7K | 5.30 | 4.31 | 3.71 | 3.46 | 3.41 |
| FFHQ 64×64 (Karras et al., 2019) | 1K | 8.63 | 7.38 | 5.65 | 4.72 | 5.45 |
| | 3K | 7.22 | 5.61 | 5.22 | 4.38 | 4.30 |
| | 5K | 7.45 | 5.82 | **4.84** | 4.37 | **3.80** |
| | 7K | **6.83** | **5.46** | **4.84** | **4.17** | 4.15 |
| ImageNet 64×64 (Deng et al., 2009) | 1K | 15.50 | 9.72 | 7.35 | 5.39 | 5.30 |
| | 3K | **11.67** | 7.87 | 5.77 | 5.15 | 4.83 |
| | 5K | 11.84 | **6.68** | 5.29 | **4.41** | 4.14 |
| | 7K | 11.80 | 6.97 | **5.11** | 4.69 | **3.97** |
| LSUN-Bedroom 256×256 (Yu et al., 2015) | 1K | 11.23 | 7.22 | 6.10 | 5.67 | 6.64 |
| | 3K | 9.59 | 6.15 | 5.31 | 4.69 | 4.72 |
| | 5K | 9.88 | 6.63 | **5.27** | **4.52** | **4.42** |
| | 7K | **8.53** | **6.14** | 5.50 | 5.11 | 4.46 |

Table 17: Evaluation on fidelity and diversity on CIFAR10 32×32 (Krizhevsky & Hinton, 2009).[†]We directly borrowed the results reported by (Zhou et al., 2024b).

| Method | NFE | FID | Precision | Recall | Density | Coverage |
|---|---|---|---|---|---|---|
| **TrajP-L (ours)** | 2 | 5.02 | 0.76 | 0.69 | 1.03 | 0.92 |
| | 3 | 3.91 | 0.78 | 0.70 | 1.07 | 0.93 |
| | 4 | 3.54 | 0.78 | 0.69 | 1.11 | 0.94 |
| | 5 | 3.34 | 0.79 | 0.70 | 1.12 | 0.94 |
| | 6 | 3.09 | 0.78 | 0.70 | 1.16 | 0.95 |
| [†]DPM-Solver++(3M) (Lu et al., 2025) | 11 | 3.93 | 0.76 | 0.71 | 1.04 | 0.94 |
| | 15 | 2.64 | 0.76 | 0.73 | 1.03 | 0.95 |
| | 19 | 2.54 | 0.77 | 0.72 | 1.04 | 0.96 |
| | 23 | 2.65 | 0.77 | 0.72 | 1.05 | 0.96 |
| | 50 | 2.01 | 0.78 | 0.72 | 1.11 | 0.96 |
| [†]DDIM (Song et al., 2020a) | 50 | 2.91 | 0.79 | 0.71 | 1.09 | 0.95 |
| [†]EDM (Karras et al., 2022) | 50 | 1.96 | 0.79 | 0.72 | 1.10 | 0.96 |

### D.2 ADDITIONAL VISUALIZE STUDY RESULTS

Additional visual results for NFE = 2 and 6 on CIFAR10 32×32 (Krizhevsky & Hinton, 2009) and ImageNet 64×64 (Deng et al., 2009) are presented in Figure 10 and Figure 12, respectively; for FFHQ 64×64 (Karras et al., 2019) and LSUN-Bedroom 256×256 (Yu et al., 2015) datasets, those with NFE = 3 and 6 are shown in Figure 11 and Figure 13, respectively. Extra visual sampling results using Stable Diffusion v1.5 (Rombach et al., 2022) are displayed in Figure 14. These visual results demonstrate that TrajP-L can generate higher-quality samples with richer details in fewer steps compared to corresponding baseline methods.

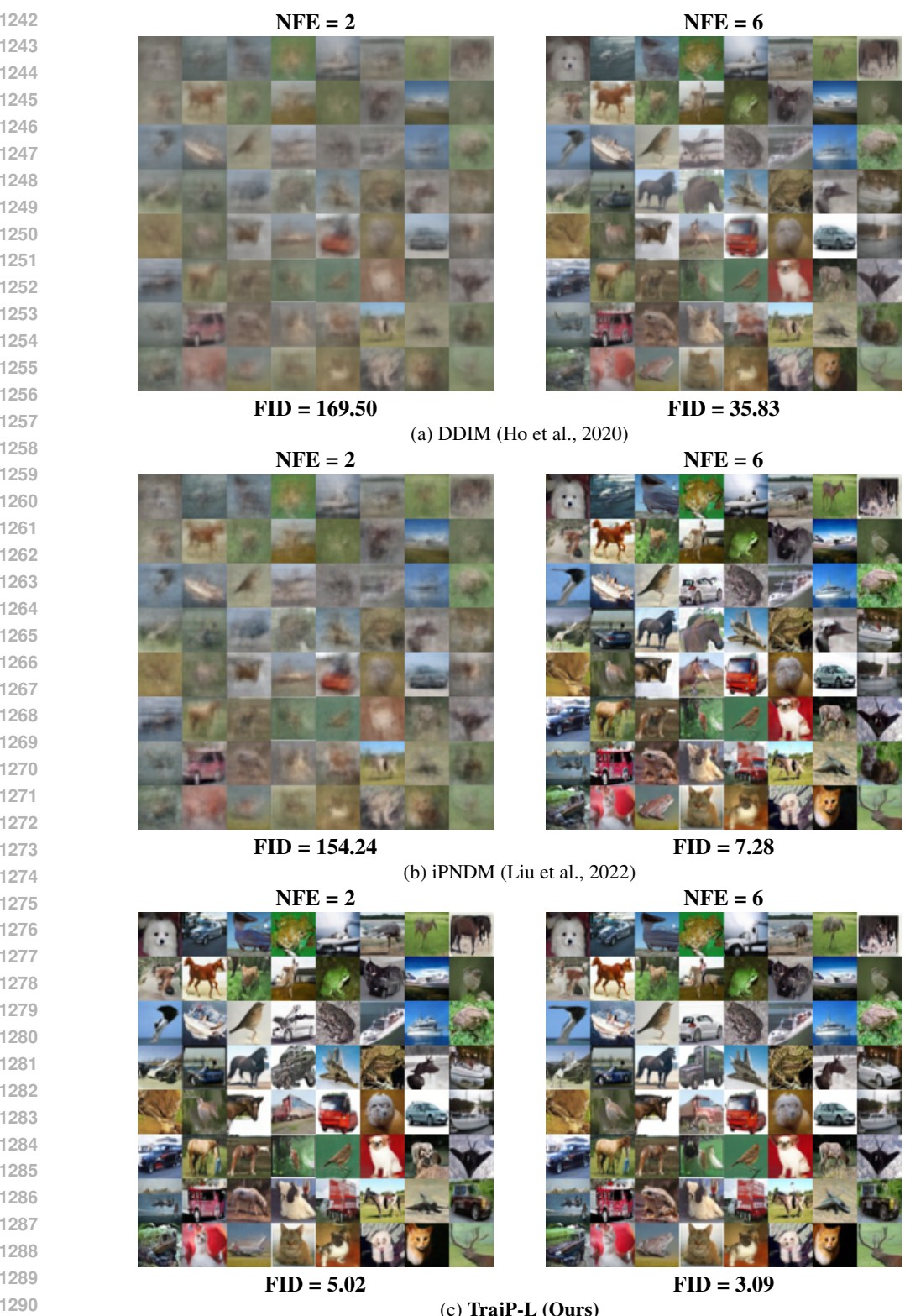

Figure 10: Samples generated on the CIFAR10 32×32 (Krizhevsky & Hinton, 2009) using the DDIM (Song et al., 2020a), iPNDM (Liu et al., 2022), and TrajP-L methods. All samples share identical generation conditions, including the use of the same random seed, adoption of a polynomial schedule time step with $\rho = 7.0$, and NFE set to 2 and 6, respectively.

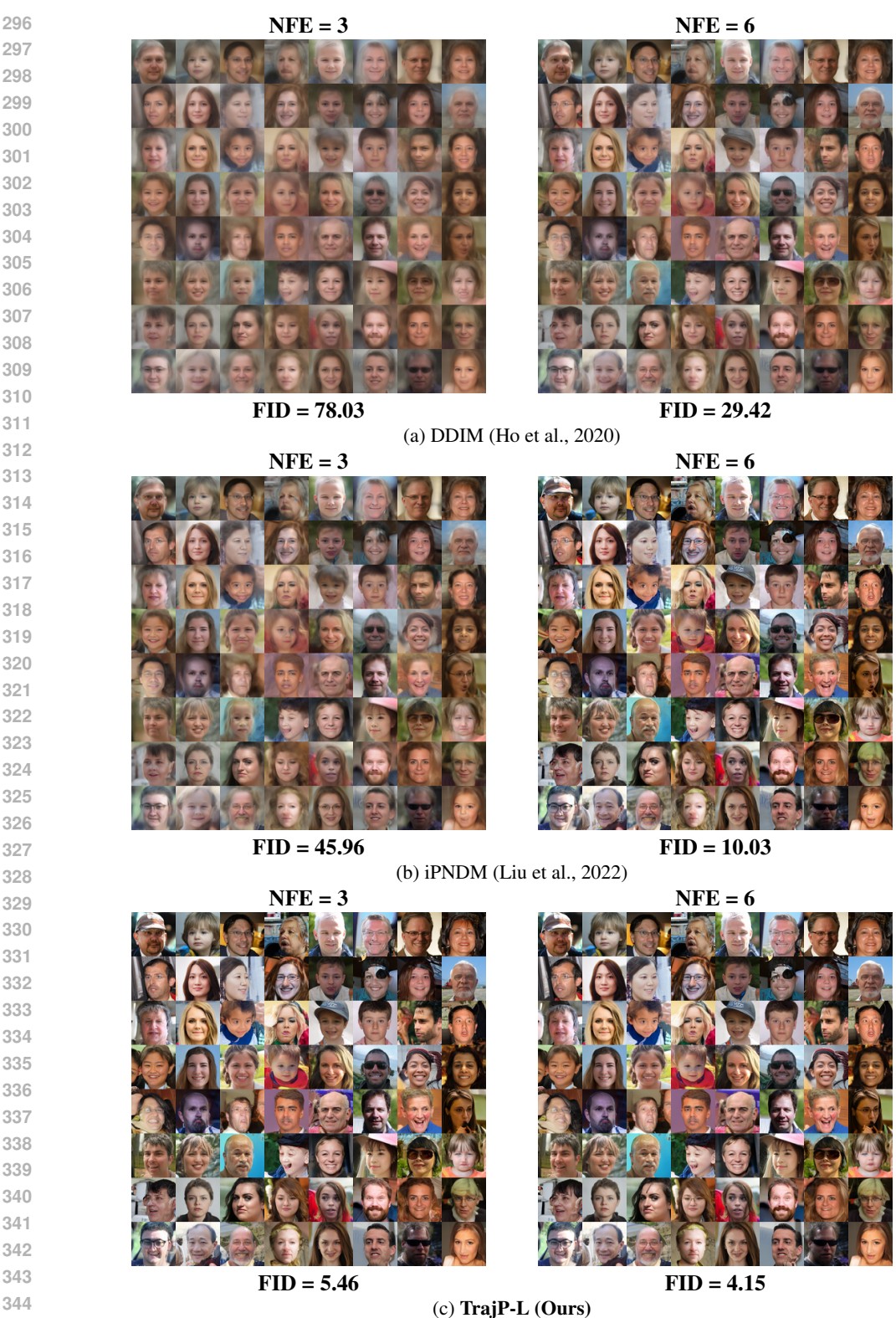

Figure 11: Samples generated on the FFHQ 64×64 (Karras et al., 2019) using the DDIM (Song et al., 2020a), iPNDM (Liu et al., 2022), and TrajP-L methods. All samples share identical generation conditions, including the use of the same random seed, adoption of a polynomial schedule time step with $\rho = 7.0$, and NFE set to 3 and 6, respectively.

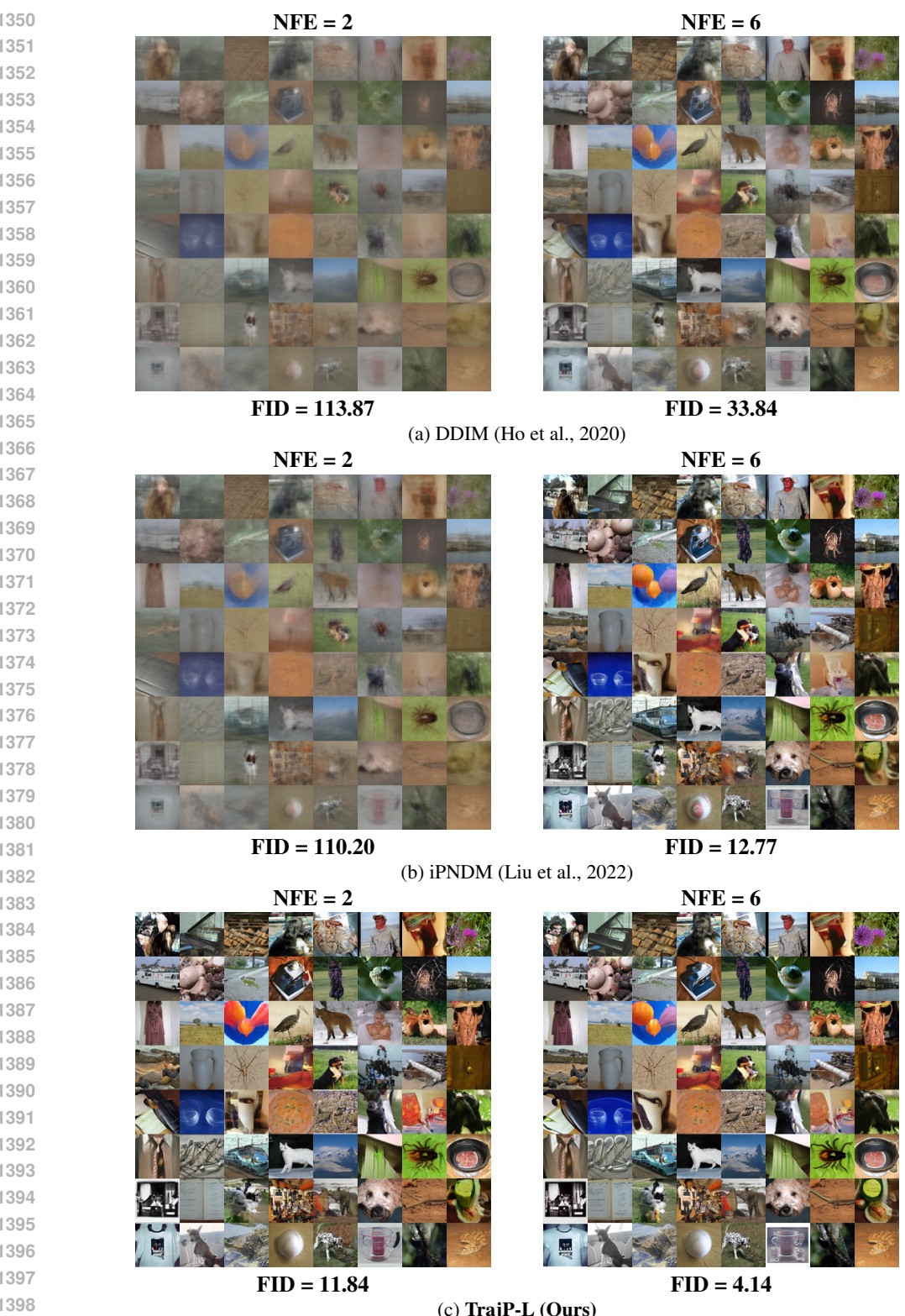

Figure 12: Samples generated on the ImageNet 64×64 (Deng et al., 2009) using the DDIM (Song et al., 2020a), iPNDM (Liu et al., 2022), and TrajP-L methods. All samples share identical generation conditions, including the use of the same random seed, adoption of a polynomial schedule time step with $\rho = 7.0$, and NFE set to 2 and 6, respectively.

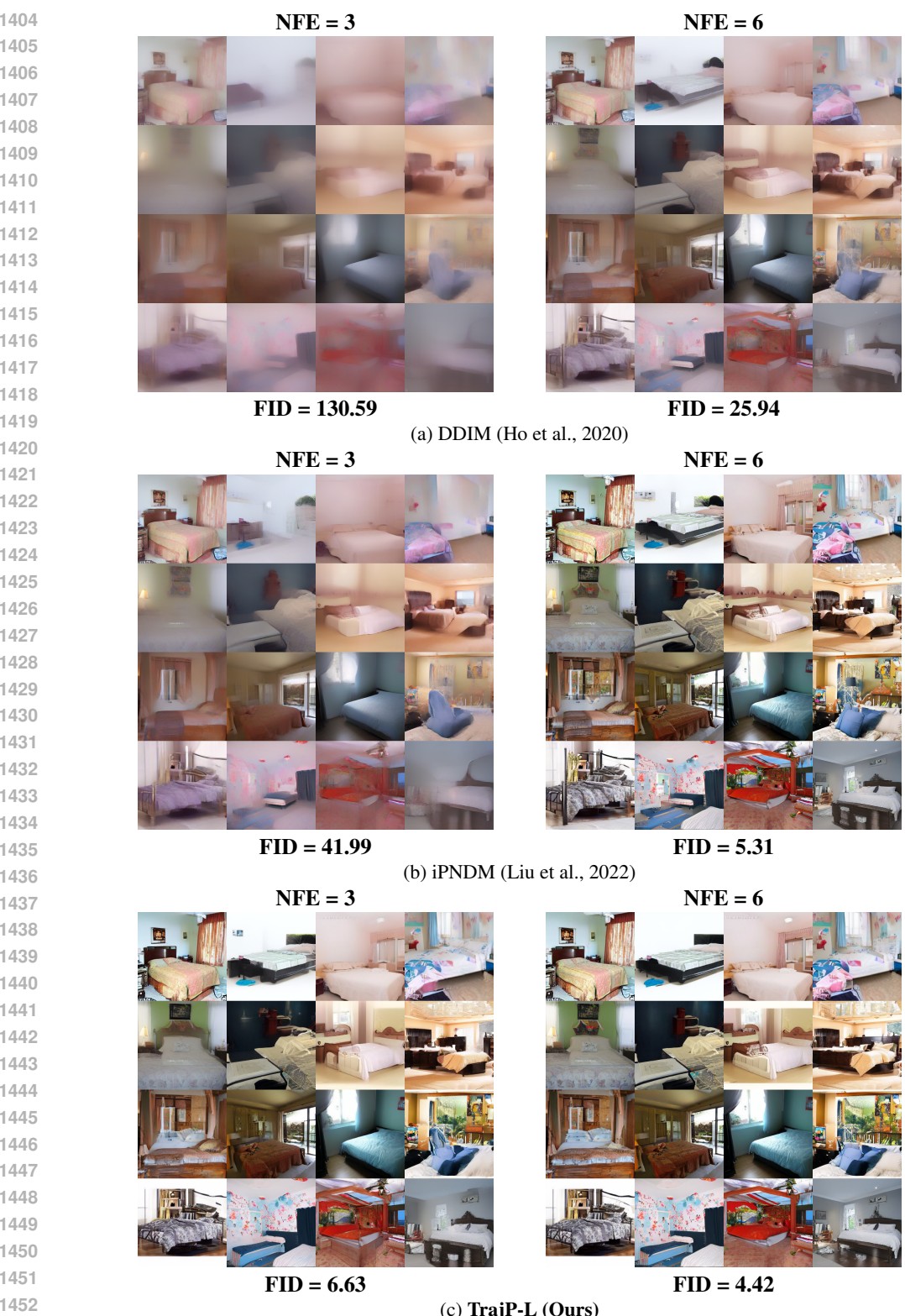

**NFE = 3**  **NFE = 6**

**FID = 130.59**  **FID = 25.94**

(a) DDIM (Ho et al., 2020)

**NFE = 3**  **NFE = 6**

**FID = 41.99**  **FID = 5.31**

(b) iPNDM (Liu et al., 2022)

**NFE = 3**  **NFE = 6**

**FID = 6.63**  **FID = 4.42**

(c) **TrajP-L (Ours)**

Figure 13: Samples generated on the LSUN-Bedroom 256×256 (Yu et al., 2015) using the DDIM (Song et al., 2020a), iPNDM (Liu et al., 2022), and TrajP-L methods. All samples share identical generation conditions, including the use of the same random seed, adoption of a uniform schedule time step, and NFE set to 3 and 6, respectively.

**Text Prompts** (listed from left to right):
    A close up of a cat on a rug on the ground.
    A clownfish swam past the coral.
    In the autumn park, the bench is covered with fallen leaves.
    A white plate is filled with various fruits.
    An erupting volcano, with the sky shrouded in dark clouds.

**NFE = 8**

**NFE = 16**

Figure 14: Samples were generated via the Stable Diffusion v1.5 (Rombach et al., 2022) using three methods: DDIM (Song et al., 2020a), DPM Solver++(2M) (Lu et al., 2025), and TrajP-L. All samples shared identical generation conditions, including a guiding scale of 7.5, the same random seed, a uniform schedule time step, and NFE set to 8 and 16, respectively.

# E  LLM USAGE STATEMENT

For this work, we leveraged a Large Language Model (LLM) as an auxiliary tool to improve the manuscript's writing and aid in producing supplementary experimental code.

