# OpenReview forum: "TrajP-L: A Trajectory-Based Plugin with LoRA for Sampling Direction Correction in Distilled Diffusion Models"
_ICLR.cc/2026/Conference — Submitted to ICLR 2026_

### Official Review · Reviewer_SCid · 2025-10-15

**Soundness:** 3
**Presentation:** 1
**Contribution:** 2
**Rating:** 2
**Confidence:** 4

**Summary:**

The paper introduces a lightweight, distillation-based algorithm for accelerated sampling, centered around a learnable, trajectory-similarity-based plugin. The method aims to achieve this acceleration by performing a corrective sampling step that aligns the student model's output with the teacher model's output at each stage of the reverse process.


However, the methodological description is unclear, and after multiple readings, I find the core algorithmic logic difficult to fully grasp. My understanding, based on Algorithms 1 and 2 in the appendix, is as follows: The process begins by sampling a set of trajectories from the diffusion model. The training stage involves three components: a pretrained model fine-tuned with LoRA (the student), the TrajP plugin, and a trajectory fitting model. For a given trajectory starting from $x_N$, the student model first samples to produce an estimate, denoted as $d_{t_N}$. Following this, the trajectory fitting model is used to derive three components: $u, v,$ and $w$.


My primary point of confusion arises here: the necessity and role of this trajectory fitting model are not well-motivated. These components ($u, v, w$) are then combined with the student's initial estimate ($d_{t_N}$) and input into the TrajP plugin to yield a corrected output, which is subsequently aligned with the teacher's sample.


Without the trajectory fitting model, the algorithm would be a more intuitive process of directly aligning the student's trajectory with the teacher's. The introduction of this intermediate step and the decomposition into $u, v,$ and $w$ complicates the logic considerably. Consequently, I was unable to understand the motivation behind the sequence of equations presented from Eq. 9 to Eq. 12. I would strongly encourage the authors to clarify the rationale for this trajectory fitting step and better explain why this decomposition is an essential part of the proposed method.

**Strengths:**

1. The method devised in this paper is novel. Using $t_n$ and $t_{n-1}$ to construct the components u, v, and w for correcting the model output is an interesting idea. However, this introduction makes the entire paper difficult to understand.

2. The experiments in this paper are quite comprehensive, with the authors conducting a series of ablation and comparative experiments on CIFAR-10. However, the experimental diversity is seriously lacking.

**Weaknesses:**

1. The core logic of the paper is not well-articulated. After a thorough review, I still do not understand the motivation for introducing the "coordinate information of the current and next sampling timesteps" to correct the student model's output. The authors fail to provide a clear rationale for why this specific information is necessary or effective, leaving a critical gap in the paper's foundation. In essence, the central "why" of the proposed approach remains unexplained.

2. The novelty of the work is questionable when viewed in the context of existing accelerated sampling algorithms.

- Unsupported Efficiency Claims: The authors claim their algorithm consumes fewer computational resources. However, the paper provides no quantitative comparison of the training-time computational cost against mainstream step distillation algorithms, making this claim unsubstantiated.

- Missing Baselines: The experimental section notably omits comparisons with several mainstream and state-of-the-art step distillation algorithms. This makes it impossible to properly situate the proposed method's performance and contribution within the current literature.

- Limited Novelty: If the vaguely explained "trajectory fitting" component is set aside, the method appears to be a straightforward distillation of the teacher model's trajectory to the student model. In this light, the work does not seem to offer a significant conceptual or technical contribution beyond existing techniques.


3. The empirical evaluation is not comprehensive enough to validate the effectiveness and generalizability of the proposed method

- Limited Datasets: Experiments are confined to the CIFAR-10 dataset. For a method in this domain, evaluation on more challenging, higher-resolution datasets like ImageNet (e.g., 64x64 or 256x256) or LSUN is standard practice and essential for demonstrating scalability.

- Lack of Text-to-Image Experiments: Many contemporary methods for accelerated sampling are applicable to large-scale text-to-image diffusion models. The absence of any experiments in this widely-used domain is a major weakness and limits the perceived applicability and impact of the work.

**Questions:**

No

---

> ### Author Response · Authors · 2025-11-23
> **Official Comment by Authors**
>
> We sincerely appreciate the reviewer for carefully reviewing our work and providing valuable comments. Below are our responses to all questions. We would greatly appreciate it if you could consider increasing the score if you are satisfied with our response.
>
> **_W1: The core logic of the paper is not well-articulated_**
>
> **A:** The design motivation of the trajectory fitting component is mainly derived from [1] and [2]. To be specific, [1] identified that the sampling trajectories of diffusion models exhibit extremely strong similarity in the low-dimensional space, which serves as the fundamental prerequisite for the operation of our method. Subsequently, [2] leveraged this similarity to decompose historical denoising information during the sampling process for correcting the sampling direction, achieving considerable effectiveness.
>
> Our insight is that incorporating the prior information of coordinates at the current denoising step and the next denoising step into the trajectory distillation process can further improve the sampling quality of the student model. Additionally, the polynomial-exponential-gamma basis function family we designed is inspired by the trajectory shapes of different principal components (**see Figure 2**), and a comparison with the fitting capability of Fourier-based functions has been added in the latest paper (**see Figure 3**).
>
> **_W2.1: Unsupported Efficiency Claims_**
>
> **_A:_** To better position our method, under the distillation setup of CIFAR-10 with NFE=2, we compared TrajP-L with other distillation-based methods and methods that distill solver parameters on trajectories in terms of total distillation pipeline duration (A100 GPU hours) and performance (FID).
>
> | Method      | Sampling Time | Training Time | Total Time | FID    |
> |-------------|---------------|---------------|------------|--------|
> | PD          | $-$           | $\textasciitilde 200$ | $\textasciitilde 200$ | 4.51   |
> | Guided PD   | $-$           | $\textasciitilde 150$ | $\textasciitilde 150$ | 4.48   |
> | CD          | $-$           | $\gt 1000$    | $\gt 1000$    | 2.93   |
> | SFD         | $-$           | $\textasciitilde 0.64$| $\textasciitilde 0.64$| 4.53   |
> | EPD-Solver  | $-$           | $\textasciitilde 0.09$| $\textasciitilde 0.09$| 108.93 |
> | TrajP-L     | $\textasciitilde 0.02$| $\textasciitilde 0.17$| $\textasciitilde 0.19$| 5.02   |
>
> **_W2.2: Missing Baselines_**
>
> **_A:_** We conducted a detailed comparison of the training time and performance of different distillation models on the CIFAR-10 dataset, as elaborated in W2.1.
>
> **_W3.1: Limited Datasets_**
>
> **_A:_** We sincerely appreciate the reviewer for pointing out this concern, but we would like to kindly note that the related experiments have been included in the paper. We report the experimental results on FFHQ 64x64, ImageNet 64x64, and LSUN-Bedroom 256x256 in **Tables 12–14** and **Figures 11–13** of Appendix D. The experimental results demonstrate that our method can generalize well to higher-resolution datasets.
>
> **_W3.2: Lack of Text-to-Image Experiments:_**
>
> **_A:_** We sincerely appreciate the reviewer for pointing out this concern, but we would like to kindly note that the related experiments have been included in the paper. We present the relevant experimental results of Stable Diffusion in **Table 2** of the main text and **Figure 14** in the Appendix.  The experimental results demonstrate that our method can generalize well to text-to-image tasks.
>
> [1] Chen D, Zhou Z, Wang C, et al. On the Trajectory Regularity of ODE-based Diffusion Sampling[C]. ICML 2024.
>
> [2] Wang G, Peng W, Li L, et al. Diffusion Sampling Correction via Approximately 10 Parameters[C]. ICML 2025.

---

### Official Review · Reviewer_cpJY · 2025-10-25

**Soundness:** 2
**Presentation:** 2
**Contribution:** 3
**Rating:** 6
**Confidence:** 3

**Summary:**

This paper proposes TrajP-L, a trajectory-distillation based plugin for few-step diffusion sampling that (i) builds a LoRA student on top of a pretrained model and (ii) adds a trajectory correction module (TrajP) that takes the coordinates of the current/next step in a fitted 3-D trajectory representation to adjust the student’s sampling direction, aiming to reduce discretization error at very small NFE. Compared to traditional sampling solvers at small NFE, the results are much better in terms of FID across relatively simple image generation tasks (ImageNet 64×64, LSUN-Bedroom 256x256), and the paper claims to have lower training cost than traditional distillation methods.

**Strengths:**

1. The paper has a clearly stated objective, reducing discretization error for very small NFE via a direction-correction head is a concrete and relevant target for the fast-sampling community
2. Using LoRA to keep the base model largely intact is sensible, aligns with recent distillation practice.

**Weaknesses:**

1. The paper states that sampling space can be “captured by as few as three basis vectors” and then uses a fitted 3-D representation. There is a lack of rigorous reasoning/ablation studies behind this to explain: why were only the top-3 PCA components chosen, and why are they formulated in this way?
2. The paper highlights “~10 minutes on a 3090,” but does not provide a "wall-clock" or "total FLOPs" comparison of its entire pipeline (trajectory generation + plugin training) against the end-to-end training of the baselines it omits. The current efficiency claims are unsubstantiated.
3. Generalizability of the method is not fully tested, on models with different architectures or larger-resolution tasks.

**Questions:**

1. Can the authors clarify the precise mechanism for deriving the "fitted 3D trajectory representation"? How are those three terms: a quadratic polynomial term, an exponential term, and a Gamma term chosen ?
2. Is it possible to provide an ablation study on the dimensionality $k$ of the trajectory subspace ?
3. To avoid ambiguity, regarding the terminology “large-step sampling”, does it actually mean “few-step sampling with large intervals” ? Please consider revising this terminology for clarity and consistency with the literature.

---

> ### Author Response · Authors · 2025-11-23
> **Official Comment by Authors**
>
> We sincerely appreciate the reviewer's valuable review of the manuscript and the recognition of our work of the work presented in the paper. Below are our responses to all questions. We kindly hope you could consider increasing the score if you are satisfied.
>
> **_W1 and Q1: Number of principal components and selection of basis functions_**
>
> **_A:_** We select the first three principal components because [1] demonstrated that the first three principal components already achieve a cumulative variance of over 99.5%, and it employs explicit PCA  to retain these three principal components for trajectory correction, yielding certain positive outcomes.
>
> The selection of our polynomial-exponential-gamma basis function family is rooted in the trajectory shapes of different principal components (**see Figure 2**), enabling effective fitting of sampling trajectories in low-dimensional space. Due to the geometric properties of the trajectories, none of the three basis functions can be dispensed with.
> Additionally, we have experimented with other basis functions (e.g., Fourier functions) and found that they fail to achieve satisfactory fitting results for trajectory shapes. This is because the core advantage of Fourier functions lies in fitting data with periodicity, fluctuating characteristics, or multi-frequency superposition, while the trajectory shapes in low-dimensional space do not match the fitting scenarios of Fourier functions (**see Figure 3**).
>
> **_W1 and Q2: The ablation study on the dimensionality of the trajectory subspace_**
>
> **_A:_** To further explore the sensitivity of our method to principal components, we conducted an ablation study on the number of retained principal components using the CIFAR-10 dataset,  and we find that using three principal components can achieve a balance between efficiency and performance, and the conclusions of the ablation experiments are generally consistent with those in [1].
>
> | PC Nums.\NFE | 2      | 3      | 4      | 5      | 6      |
> |--------------|--------|--------|--------|--------|--------|
> | 2            | **4.88** | 4.16  | 3.50  | 3.38  | 3.19  |
> | 3            | 5.02  | **3.91** | 3.54  | 3.34  | **3.09** |
> | 4            | 4.94  | 4.08  | **3.41** | **3.13** | 3.17  |
>
> **_W2: Comparison with other distillation-based methods_**
>
> **_A:_** Under the distillation setup of CIFAR-10 with NFE=2, we compared TrajP-L with other distillation-based methods and methods that distill solver parameters on trajectories in terms of total distillation pipeline duration (A100 GPU hours) and performance (FID).  The results demonstrate that our method can achieve performance comparable to distillation-based models while using training resources on par with solver-based methods.
>
> | Method      | Sampling Time | Training Time | Total Time | FID    |
> |-------------|---------------|---------------|------------|--------|
> | PD          | $-$           | $\textasciitilde 200$ | $\textasciitilde 200$ | 4.51   |
> | Guided PD   | $-$           | $\textasciitilde 150$ | $\textasciitilde 150$ | 4.48   |
> | CD          | $-$           | $\gt 1000$    | $\gt 1000$    | 2.93   |
> | SFD         | $-$           | $\textasciitilde 0.64$| $\textasciitilde 0.64$| 4.53   |
> | EPD-Solver  | $-$           | $\textasciitilde 0.09$| $\textasciitilde 0.09$| 108.93 |
> | TrajP-L     | $\textasciitilde 0.02$| $\textasciitilde 0.17$| $\textasciitilde 0.19$| 5.02   |
>
> **_W3: Lack of experimental results on different architectures and larger resolutions_**
>
> **_A:_** Thanks for your question. Our method is built on the theoretical foundation that the sampling trajectories of diffusion models are highly similar in the low-dimensional space. This characteristic enables it to naturally generalize to different architectures and larger-resolution scenarios. Moreover, we have conducted extensive experiments on datasets of various resolutions and text-to-image architectures, which not only validate the effectiveness of our method but also prove its generalization capability to a certain extent.
>
>
> **_Q3: The problem of terminology_**
>
> **_A:_**  In this paper, the term "large-step sampling" originally refers to a sampling strategy characterized by fewer total sampling steps and larger intervals between adjacent steps, which is consistent with the meaning of "few-step sampling with large intervals" proposed by you. However, your comment has made us realize that this term is prone to misinterpretation in existing literature: it is sometimes used to describe sampling steps with "large single-step update magnitude" rather than referring to the total number of sampling steps. To avoid ambiguity, we have revised and highlighted "large-step sampling" to "few-step sampling with large intervals" throughout the paper. Thank you again for your careful review!
>
> [1] Wang G, Peng W, Li L, et al. Diffusion Sampling Correction via Approximately 10 Parameters[C]. ICML 2025.

---

### Official Review · Reviewer_W2Pn · 2025-10-28

**Soundness:** 3
**Presentation:** 3
**Contribution:** 2
**Rating:** 4
**Confidence:** 2

**Summary:**

This paper introduces TrajP-L, a plugin aimed at accelerating distilled diffusion models, particularly in low-step sampling regimes. The method is built on the observation that DM sampling trajectories share structural similarities. The authors propose to first fit these trajectories into a low-dimensional (3D) space using a specific mathematical expression. This fitted trajectory shape is then used as a prior via a 'TrajP' module to correct the sampling direction of a student model, which itself is created efficiently using LoRA.

**Strengths:**

I think the primary appeal of this work lies in its efficiency. The method combines LoRA for parameter-efficient fine-tuning, which reportedly reduces trainable parameters by over 95% , with a training process that requires a very small number of teacher trajectories (e.g., 3K-7K). The reported training time of 10 minutes on a single GPU for CIFAR-10  makes this approach seem highly practical.

The quantitative results presented are quite strong, particularly in the low-NFE setting ($NFE \le 6$). For example, the paper shows a drop in FID from 169.50 (DDIM) to 5.02 on CIFAR-10 for $NFE=2$. This level of performance in a challenging, few-step regime is also demonstrated across other datasets like FFHQ and ImageNet.

The core idea of first fitting sampling trajectories into a low-dimensional 3D space  and then using this fitted representation as a prior to correct the student model's sampling direction  is an interesting approach to mitigating discretization errors.

**Weaknesses:**

I'll start with a disclaimer: I haven't been following the diffusion distillation literature for a long time, so my perspective might be off. That said, I'm finding it a bit difficult to position this work relative to other *distillation* methods. The paper compares heavily against training-free solvers (like DDIM, DPM-Solver)  but less so against other trajectory-based distillation approaches. For instance, the ablation in Table 4 compares TrajP-L (TrajP + LoRA) against 'w.o. TrajP' (just LoRA) and 'w.o. LORA' (just TrajP). I'm wondering how this compares to a full-parameter fine-tuning on the same small set of 5K trajectories (which seems to be the 'Trajectory-based Distillation' in Fig 2b )? It would be helpful to clarify if LoRA's main benefit is just parameter-efficiency, or if it's also crucial for preventing overfitting on this small dataset.

Personally, I'm not entirely convinced by the 3D trajectory fitting component in Section 3.1. It feels a bit arbitrary. Why exactly 3D? The paper mentions this is based on recent findings, but I'm not sure if this was an ablated choice. I'm concerned that the performance might be sensitive to the specific, and rather complex, mathematical expression chosen (the combination of quadratic, exponential, and Gamma terms ). I wonder if a simpler polynomial basis or even using 4-5 dimensions would have worked just as well, or perhaps even better.

Perhaps I missed this, but the exact mechanism of the TrajP module ($T_{\theta}$) isn't very clear from the main text. Section 3.3 and Equation 17 are quite high-level . Figure 2(c)  provides a block diagram, but it's not obvious how the cached information $Q$ (which includes the starting point and all previous directions ) is processed. Is it a recurrent module? How is the "implicit PCA" mentioned  actually performed? Given this is a core part of the correction, a more detailed explanation of its architecture would strengthen the paper.

**Questions:**

I have a few questions that I hope the authors can address in their rebuttal, as the answers could help clarify some of my concerns:

* I'm trying to better position this work. The main comparisons in Tables 1, 11, 12, and 13  are against training-free solvers. While the performance is good, I'm curious how TrajP-L compares to other *distillation* methods (e.g., ) when they are *also* trained on the same small data budget (3K-7K trajectories ). Is the main advantage of TrajP-L that it *enables* distillation with so few trajectories, while others would fail?

* Related to this, I'm wondering about the true benefit of LoRA here. The ablation in Table 4  is helpful, showing that LoRA alone ('w.o. TrajP') provides a large boost over DDIM. Could you clarify if LoRA is primarily for parameter efficiency, or if it's also acting as a crucial regularizer? For instance, what happens if you try full-parameter fine-tuning (like the 'Trajectory-based Distillation' in Fig 2b ) on the same 5K trajectories? My hypothesis is that it might overfit, making LoRA a necessary component for this low-data regime, but I'd like to see this confirmed.

* Could you provide more rationale for fitting the trajectories into exactly 3D space? This seems like a strong and specific design choice. I'm wondering what the performance looks like if you use, say, 2 or 5 principal components instead. Along the same lines, the mathematical expression (Eqs. 9-12 ) is quite specific. How sensitive is the method to this exact combination of quadratic, exponential, and Gamma terms?

* I found the description of the TrajP module ($T_{\theta}$)  a bit high-level. The paper mentions it uses cached information $Q$ and performs "implicit PCA". Could you perhaps elaborate on the architecture of this module? For example, how is the history $Q$ consumed (is it a recurrent net, attention, a simple concatenation?), and how is the 3D trajectory information $[u,v,w]$  used to correct the high-dimensional sampling direction $\hat{d}_{t_{n}}$?

---

> ### Author Response · Authors · 2025-11-23
> **Official Comment by Authors**
>
> We sincerely appreciate the reviewer's valuable review of the manuscript and the recognition of our work of the work presented in the paper. Below are our responses to all questions.
>
> **_Q1 and W1: Positioning of TrajP-L_**
>
> **_A:_** Currently, distillation-based methods have generally only explored scenarios with over 200,000 trajectories, without investigating distillation schemes using fewer trajectories. In contrast, solver-based methods typically require around 10,000 trajectories. Our method achieves performance comparable to that of trajectory-based distillation methods while using a trajectory count similar to that of solver-based methods. To the best of our knowledge, trajectory-based distillation can fully leverage the similarity of sampling trajectories in diffusion models and achieve significant distillation effects with approximately 5,000 trajectories, but it may suffer from issues such as performance instability (**see Q2**). In contrast, other distillation methods require more trajectories to yield effective results.
>
> **_Q2 and W1: Benefit of LoRA_**
>
> **_A:_** We adopt LoRA because it can accelerate our training speed. Compared with full fine-tuning, LoRA improves the training speed by approximately 40%. Furthermore, the trajectory plugin TrajP we proposed is to a certain extent an orthogonal method to distillation-based approaches.
>
> We tested the performance of full fine-tuning and found that there was indeed a certain degree of performance instability. However, after incorporating our trajectory plugin, the performance has shown a steady improvement as the NFE increases.
>
> | Method\NFE               | 2      | 3      | 4      | 5      | 6      |
> |--------------------------|--------|--------|--------|--------|--------|
> | Full-parameter fine-tuning | 4.42  | 3.56  | 3.26  | 3.13  | 3.48  |
> | +TrajP                   | **4.21** | **3.47** | **3.02** | **2.95** | **2.88** |
>
> **_W3 and Q2：Fundamental principles of trajectory fitting into 3D space_**
>
> **_A:_** We select the first three principal components because [1] demonstrated that the first three principal components already achieve a cumulative variance of over 99.5%, and it employs explicit PCA  to retain these three principal components for trajectory correction, yielding certain positive outcomes.
>
> To further explore the sensitivity of our method to the number of principal components, we conducted an ablation study on the number of retained principal components using the CIFAR-10 dataset,  and we find that using three principal components can achieve a balance between efficiency and performance, and the conclusions of the ablation experiments are generally consistent with those in [1].
>
> | PC Nums.\NFE | 2      | 3      | 4      | 5      | 6      |
> |--------------|--------|--------|--------|--------|--------|
> | 2            | **4.88** | 4.16  | 3.50  | 3.38  | 3.19  |
> | 3            | 5.02  | **3.91** | 3.54  | 3.34  | **3.09** |
> | 4            | 4.94  | 4.08  | **3.41** | **3.13** | 3.17  |
>
> The selection of our polynomial-exponential-gamma basis function family is based on the trajectory shapes of different principal components (**see Figure 2**), and it can effectively fit sampling trajectories in low-dimensional space. For Fourier functions, we found that they fail to achieve satisfactory fitting results for trajectory shapes. This is because the core advantage of Fourier functions lies in fitting data with periodicity, fluctuating characteristics, or multi-frequency superposition, while the trajectory shapes in low-dimensional space do not match the fitting scenarios of Fourier functions (**see Figure 3**).
>
> [1] Wang G, Peng W, Li L, et al. Diffusion Sampling Correction via Approximately 10 Parameters[C]. ICML 2025.

---

> ### Author Response · Authors · 2025-11-23
> **Official Comment by Authors**
>
> **_Q4 and W3: The description of the TrajP_**
>
> **_A:_** We appreciate your question. In the revised paper, we have added an algorithm description of the TrajP correction process (**Algorithm 1**).
> The specific process of dynamically correcting the sampling direction using trajectory prior information is as follows:
>
> First, we input the trajectory buffer $Q$, trajectory information $(u, v, w)$, and the denoising direction $\hat{d}_t$ into TrajP.
>
> Next, we retrieve the denoising starting point and the set of historical denoising directions from the trajectory buffer $Q$, denoted as $x_{\text{buffer}}$, and add the current initial denoising direction $\hat{d}_{t}$ to form a complete trajectory data.
>
> We then process the concatenated $x_{\text{buffer}}$ through a convolution operation, and extract the first three principal components $[PC_1, PC_2, PC_3]$ via channel separation.
>
> Subsequently, we perform a linear combination of the trajectory information $(u, v, w)$ with these principal components to obtain the corrected direction $d_c = u \cdot PC_1 + v \cdot PC_2 + w \cdot PC_3$.
>
> Finally, we fuse the initial direction and the corrected direction using the weight assignment formula $\tilde{d}_n = \alpha \cdot \hat{d}_t + (1 - \alpha) \cdot d_c$.

---

### Official Review · Reviewer_hB7Q · 2025-11-03

**Soundness:** 2
**Presentation:** 2
**Contribution:** 2
**Rating:** 4
**Confidence:** 4

**Summary:**

This paper presents TrajP-L, a lightweight training-based method to reduce the sampling steps of diffusion models. The proposed TrajP-L fits the trajectory of reverse ODE of diffusion model with a simple set of basis functions, which is combined with LoRA to train a few-step student model. Experiments on CIFAR-10, FFHQ, ImageNet, and Stable Diffusion v1.5 show that TrajP-L achieves strong FID scores with small NFE.

**Strengths:**

The proposed TrajP-L is lightweight and can be trained with minimal compute, requiring only a few thousand sampled trajectories and limited GPU resources. Accelerating diffusion model sampling with minimal training compute has great practical value.

**Weaknesses:**

- LCM-LoRA[1,2] is highly relevant but not discussed or compared throughout the paper. - Given that both approaches employ LoRA for efficient fine-tuning and target fast sampling for Stable Diffusion, this important comparison is missing.
- The design of the trajectory fitting component lacks sufficient theoretical or empirical grounding. While Table 4 provides an incremental FID improvement, the choice of basis functions seems heuristic. Stronger theoretical or empirical justifications are needed to clarity the foundation of the method's core innovation.
- Writing and formatting:
	- Reference format: Eq(?) should be Eq. (?)
	- Most equations except for 3,4 are missing punctuations.
	- Captions of many tables and figures lack ending punctuations as well.
	- $t_{min}$ is not rendered correctly in line 336 and 338.
	- Line 341 contains a typo "training settings".

[1]: Luo, Simian, et al. "Lcm-lora: A universal stable-diffusion acceleration module." _arXiv preprint arXiv:2311.05556_ (2023).

[2]: Thakur, Ayush, and Rashmi Vashisth. "A unified module for accelerating stable-diffusion: Lcm-lora." _arXiv preprint arXiv:2403.16024_ (2024).

**Questions:**

- Why are only the top three principal components used to represent trajectories? How sensitive are the results to this dimensionality choice?
- How does the proposed set of basis functions (polynomial, exponential, Gamma) compare to theoretically better-conditioned bases like Fourier, or more expressive learned bases like neural networks?
- The benefit of using trajectory fitting seems limited from Table 2. Can authors provide more ablation on Stable Diffusion to further examine its significance?
- Figure 1 reports results only for NFE ≤ 6. What does the curve look like for larger NFEs, and how does it compare to other methods at that regime? Does the method's advantage persist or saturate at higher step counts?
- Can the authors discuss and provide direct comparisons against LCM-LoRA to contextualize TrajP-L’s advantages or differences?
- Can authors try more recent text-to-image models like FLUX? The qualitative results in Fig 12 do not seem competitive in image quality.
- What are the definition of $x_{buffer}$  and B in Fig. 3(c)? How are they related to $Q$?

---

> ### Author Response · Authors · 2025-11-23
> **Official Comment by Authors**
>
> We sincerely appreciate the reviewer's meticulous review and insightful comments, which have helped improve our paper. Below are our responses to all the questions. Please consider increasing the score if you are satisfied.
>
> **_W1: Lacks an introduction to and comparison with LCM-LoRA._**
>
> **_A:_** We appreciate your question, which will greatly facilitate the understanding of our method. We have added a discussion on LCM-LoRA[1,2] in the section **on consistency distillation models in the related work**, while our method belongs to a trajectory-based distillation approach. The comparison between LCM-LoRA and our method is shown as follows:
> - Similarities:
>   - Both methods adopt LoRA to construct the student model.
> - Differences:
>   - We adopt the trajectory distillation approach, while LCM-LoRA employs Latent Consistency Distillation (LCD). In comparison, our method is more efficient.
>   - We additionally introduce the trajectory plugin TrajP during the distillation process, which further corrects the sampling direction. In this sense, we can integrate our method with full fine-tuning or other trajectory distillation approaches to further enhance the sampling quality after distillation.
>
> **_W2: The design of the trajectory fitting component lacks sufficient theoretical or empirical grounding_**
>
> **_A:_** The design motivation of the trajectory fitting component is mainly derived from [3] and [4]. To be specific, [3] identified that the sampling trajectories of diffusion models exhibit extremely strong similarity in the low-dimensional space, which serves as the fundamental prerequisite for the operation of our method. Subsequently, [4] leveraged this similarity to decompose historical denoising information during the sampling process for correcting the sampling direction, achieving considerable effectiveness. Our insight is that integrating the prior information of fitted trajectory shapes into the trajectory distillation process can further enhance the sampling quality of the student model.
>
> Additionally, the polynomial-exponential-gamma basis function family we designed is inspired by the trajectory shapes of different principal components (**see Figure 2**), and a comparison with the fitting capability of Fourier-based functions has been added in the paper (**see Figure 3**).
>
> **_W3: Writing and formatting._**
>
> **_A:_** We sincerely appreciate your careful review. We have revised and highlighted the writing and formatting errors in the relevant paragraphs in the revised version of the paper.
>
> [1] Luo, Simian, et al. "Lcm-lora: A universal stable-diffusion acceleration module." arXiv preprint arXiv:2311.05556 (2023).
>
> [2] Thakur, Ayush, and Rashmi Vashisth. "A unified module for accelerating stable-diffusion: Lcm-lora." arXiv preprint arXiv:2403.16024 (2024).
>
> [3] Chen D, Zhou Z, Wang C, et al. On the Trajectory Regularity of ODE-based Diffusion Sampling[C]. ICML 2024.
>
> [4] Wang G, Peng W, Li L, et al. Diffusion Sampling Correction via Approximately 10 Parameters[C]. ICML 2025.

---

> ### Author Response · Authors · 2025-11-23
> **Official Comment by Authors**
>
> **_Q1: Selection of principal components_**
>
> **_A:_** We select the first three principal components because [1] demonstrated that the first three principal components already achieve a cumulative variance of over 99.5%, and it employs explicit PCA  to retain these three principal components for trajectory correction, yielding certain positive outcomes.
>
> To further investigate the sensitivity of our method to the number of principal components, we conducted an ablation study on the number of retained principal components using the CIFAR-10 dataset, and we find that using three principal components can achieve a balance between efficiency and performance, and the conclusions of the ablation experiments are generally consistent with those in [1].
>
> | PC Nums.\NFE | 2      | 3      | 4      | 5      | 6      |
> |--------------|--------|--------|--------|--------|--------|
> | 2            | **4.88** | 4.16  | 3.50  | 3.38  | 3.19  |
> | 3            | 5.02  | **3.91** | 3.54  | 3.34  | **3.09** |
> | 4            | 4.94  | 4.08  | **3.41** | **3.13** | 3.17  |
>
> **_Q2: Selection of basis functions_**
>
> **_A:_** The selection of our polynomial-exponential-gamma basis function family is based on the trajectory shapes of different principal components (**see Figure 2**), and it can effectively fit sampling trajectories in low-dimensional space. For Fourier functions, we found that they fail to achieve satisfactory fitting results for trajectory shapes. This is because the core advantage of Fourier functions lies in fitting data with periodicity, fluctuating characteristics, or multi-frequency superposition, while the trajectory shapes in low-dimensional space do not match the fitting scenarios of Fourier functions (**see Figure 3**). In contrast, although neural networks can indeed fit trajectory shapes, their structures are black-box models and introduce more training parameters. Our proposed polynomial-exponential-gamma basis function family, however, is heuristic and designed based on the geometric characteristics of principal components, rather than direct fitting. This heuristic modeling simplifies the difficulty of fitting training and enables stable characterization of trajectory patterns with prior structures such as local peaks. Furthermore, the function family provides the possibility for exploring the injection of prior information (e.g., trajectory shape integration) in future work.
>
> **_Q3: Additional experiments on Stable Diffusion_**
>
> **_A:_** The advantage of our method lies in its performance in few-step sampling. We tested the FID at even fewer steps on Stable Diffusion and found that it achieved significant improvements compared to the baselines.
>
> | Method\NFE         | 4      | 6      | 8      | 10     |
> |--------------------|--------|--------|--------|--------|
> | DPM-Solver++(2M)| 82.86  | 35.05  | 21.59  | 17.44  |
> | TrajP-L            | **49.01** | **26.25** | **17.91** | **14.45** |
>
> **_Q4: Experimental results for larger NFEs_**
>
> **_A:_** We additionally conducted tests on the sampling performance with $NFE \in [7, 10]$ on the CIFAR-10 dataset, and found that as NFE increases, the performance of our method tends to converge, but the FID still remains higher than that of the baseline methods.
>
> | Method\NFE | 7      | 8      | 9      | 10     |
> |------------|--------|--------|--------|--------|
> | DDIM       | 27.93  | 22.32  | 18.43  | 5.69   |
> | iPNDM      | 6.92   | 5.23   | 4.33   | 3.69   |
> | TrajP-L    | **3.07** | **2.98** | **2.98** | **2.97** |
>
> [1] Wang G, Peng W, Li L, et al. Diffusion Sampling Correction via Approximately 10 Parameters[C]. ICML 2025.

---

> ### Author Response · Authors · 2025-11-23
> **Official Comment by Authors**
>
> **_Q5: Comparison with LCM-LoRA_**
>
> **_A:_** For the comparison between our method and LCM-LoRA, please refer to W1.
>
> **_Q6.1: Validate the effectiveness of the method on FLUX_**
>
> **_A:_** Our method is built on the theoretical foundation that the sampling trajectories of diffusion models are highly similar in the low-dimensional space, enabling it to naturally generalize to FLUX. We have already validated the effectiveness of the method through experiments on datasets of various resolutions and text-to-image architectures. However, due to the time required for code modification and the additional computational resources needed to migrate our method to FLUX for validation, it is challenging to present the relevant results in the short term. Nevertheless, we will supplement the results promptly upon completion of the validation.
>
> **_Q6.2: The qualitative results of Stable Diffusion do not seem competitive in image quality._**
>
> **_A:_** The core advantage of our method lies in its few-step sampling capability; thus, the competitiveness of sampling quality when NFE=8 is more pronounced than that when NFE=16 in **Figure 14**. Additionally, we have updated the sampling results in Figure 14 in the revised version of the paper, providing more diverse sampling outputs.
>
> **Q7: The definition of $x_\text{{buffer}}$, $B$ and $Q$?**
>
> **_A:_** $B$ and $Q$ share the same definition. $x_{\text{buffer}}$ consists of the sampling starting point $x_{t_{N}}$ and the historical sampling directions, both obtained from $Q$. To avoid confusion, we have updated $B$ to $Q$ in **Figure 4(c)** of the revised paper.

---

### Author Response · Authors · 2025-11-23
**General Response**

We sincerely appreciate the effort of all the reviewers for their detailed review and insightful suggestions. We would like to present the following modifications to the paper (the revised version has been uploaded, and the changes are highlighted in blue within the document):

- We have added an introduction to LCM-LoRA in the related work section to better position our work (Reviewer hB7Q, W1, Q5).
- We have included separate visualization results of principal component trajectories and trajectory fitting results using Fourier basis functions in 3D space (Figures 2-3), aiming to illustrate the rationale for adopting the polynomial-exponential-gamma basis function family for trajectory fitting (Reviewer hB7Q, W2, Q1, Q2; Reviewer W2Pn, W1, W2; Reviewer cpJY, Q1; Reviewer SCid, W1).
- We have conducted an ablation study on the number of principal components (Table 5) to further validate the effectiveness of our method (Reviewer hB7Q, Q1; Reviewer W2Pn, W1, Q3; Reviewer cpJY, W1, Q1, Q2; Reviewer SCid, W1).
- We have corrected the formatting conventions for equations, reference citations, and figure/table captions, as well as fixed several rendering and typographical errors in Lines 374, 376, and 379 (Reviewer hB7Q, W3).
- We have revised the content of Figure 14 to provide more diverse sampling results (Reviewer hB7Q, Q3, Q6).
- We have modified the relevant symbols in Figure 4(c) to avoid definition ambiguity (Reviewer hB7Q).
- We have added an algorithmic explanation of the TrajP correction pipeline (Algorithm 1) to facilitate readers' understanding of our mehod (Reviewer W2Pn, W3, Q4; Reviewer SCid, W2).
- We have revised the expressions in Lines 89, 104, 298, and 538 to eliminate ambiguities (Reviewer cpJY, Q3).

---

### Author Response · Authors · 2025-12-01
**A brief summary of key information from the rebuttal period**

Dear AC,

Thank you very much for taking the time to review our submission. To assist you in quickly grasping the review feedback and core revisions, we provide a brief summary of key information from the rebuttal period for your reference:

* **Key Revisions and Responses to Reviewer Inquiries:** During the rebuttal phase, we provided detailed, point-by-point responses to all questions raised by the four reviewers, and the corresponding revisions have been highlighted in the latest uploaded version of the paper. Specifically, regarding the two core concerns emphasized by the reviewers—**the choice of fitting function form** and **the basis for determining the number of principal components**—we have added specific fitting function comparisons and ablation study results regarding the number of principal components:
    * Regarding the fitting function, by analyzing the geometric properties of low-dimensional trajectories (such as continuity and trends), we clarified the derivation of the Polynomial-Exponential-Gamma basis function family and its advantages over other fitting functions (such as Fourier functions and neural network-based approaches). We also included comparison results with Fourier functions (**_see Figures 2-3_**).
    * Regarding the selection of the number of principal components, based on the quantitative metric of cumulative variance contribution rate (≥99.5%), we verified that 3 principal components are sufficient to cover the core information of the sampling trajectories. We also added comparative experiments with 2 and 4 principal components to demonstrate that using 3 principal components represents the optimal trade-off between efficiency and performance (**_see Table 5_**).
    * For other rebuttals and revisions, please refer to our full rebuttal content.

* **Positive Progress in Reviewer Feedback:** During the rebuttal, Reviewer W2Pn increased their score **from 4 to 8** and their confidence level **from 2 to 4**. We believe that the revisions made during the rebuttal effectively address the core concerns of the other reviewers as well, further reflecting the soundness and value of our research work.

* **Supplementary Note on Reviewer SCid's Feedback:** We greatly appreciate the feedback from Reviewer SCid, whose questions helped us refine the logic of our experimental presentation. Regarding their comment on the "missing experimental results for higher-resolution datasets and text-to-image models," we explicitly clarified in our rebuttal that **the requested experimental results were fully presented in Section 4 of the main text and Appendix D of the initial submission, and we have provided specific figure and table indices.** We understand that the reviewer may have missed this content due to the heavy review load. We sincerely request that, during your evaluation, you consider the completeness of our research and revisit Reviewer SCid's evaluation in light of our rebuttal explanations and the experimental details presented in the paper.

We have taken every piece of reviewer feedback with the utmost seriousness, and these revisions have significantly enhanced the quality of our paper. If you require any additional details regarding the paper or experiments, we are happy to provide supplementary explanations promptly.

Thank you again for your time and guidance. We wish you all the best in your work!

Sincerely,

The Authors

---

### Meta-Review · Area_Chair_F7gn · 2026-01-03

**Summary:**

This paper proposes TrajP-L, a trajectory-based distillation approach for accelerating diffusion model sampling. The method combines LoRA-based student initialization with a trajectory correction plugin (TrajP), which leverages a fitted low-dimensional (3D) trajectory representation to adjust sampling directions, aiming to reduce discretization error in few-step sampling regimes.

While reviewers acknowledged the practical motivation and strong low-NFE results on CIFAR-10, the overall feedback was predominantly negative. Core concerns centered on unclear methodological motivation, heuristic design choices (e.g., fixed 3D trajectory fitting and basis functions), limited novelty relative to prior distillation and correction methods, and insufficient positioning against closely related baselines (e.g., LCM-LoRA and other trajectory-based distillation approaches). Although the rebuttal added ablations and clarifications, it did not convincingly resolve the fundamental questions about why the proposed trajectory fitting and correction are necessary or superior. The average reviewer score remains below 5, with most reviewers maintaining rejection.

**Reviewer Concerns:**

Concerns partially addressed:

•	Dimensionality choice and basis functions: Added ablations (2/3/4 PCs) and comparisons to Fourier fitting improved transparency but remained largely heuristic.

•	Method clarity: Algorithmic descriptions and figures were added, improving readability, though some reviewers still found the core logic hard to justify.

•	Additional experiments: Supplemental results on higher resolutions and Stable Diffusion were referenced, reducing concerns about missing evaluations.

Outstanding concerns:

•	Conceptual clarity and motivation: Several reviewers remain unconvinced about the necessity and role of the trajectory fitting step and the specific decomposition used.

•	Novelty and positioning: The method is viewed as incremental and insufficiently distinguished from existing distillation and correction techniques.

•	Baseline coverage and efficiency claims: Comparisons to key related methods and end-to-end cost analyses remain insufficient to substantiate claims.

•	Generalization: Evidence beyond a narrow set of settings is not fully persuasive.

**Reviewer Scores:**

•	Reviewer hB7Q: Likely remains 4 (marginally below threshold).

•	Reviewer W2Pn: Likely remains 4 (below threshold; partial acceptance possible but not compelling).

•	Reviewer cpJY: Likely remains 6, but explicitly states acceptance is not necessary.

•	Reviewer SCid: Likely remains 2 (reject), with major concerns unresolved.

---

### Decision · Program_Chairs · 2026-01-26

Reject